# How many neurons are sufficient for perception of cortical activity?

**Henry WP Dalgleish[†‡], Lloyd E Russell[†], Adam M Packer[†§], Arnd Roth, Oliver M Gauld, Francesca Greenstreet[‡], Emmett J Thompson[‡], Michael Häusser***

Wolfson Institute for Biomedical Research, University College London, London, United Kingdom

**Abstract** Many theories of brain function propose that activity in sparse subsets of neurons underlies perception and action. To place a lower bound on the amount of neural activity that can be perceived, we used an all-optical approach to drive behaviour with targeted two-photon optogenetic activation of small ensembles of L2/3 pyramidal neurons in mouse barrel cortex while simultaneously recording local network activity with two-photon calcium imaging. By precisely titrating the number of neurons stimulated, we demonstrate that the lower bound for perception of cortical activity is ~14 pyramidal neurons. We find a steep sigmoidal relationship between the number of activated neurons and behaviour, saturating at only ~37 neurons, and show this relationship can shift with learning. Furthermore, activation of ensembles is balanced by inhibition of neighbouring neurons. This surprising perceptual sensitivity in the face of potent network suppression supports the sparse coding hypothesis, and suggests that cortical perception balances a trade-off between minimizing the impact of noise while efficiently detecting relevant signals.

**\*For correspondence:**
m.hausser@ucl.ac.uk

[†]These authors contributed equally to this work

**Present address:** [‡]UCL Sainsbury Wellcome Centre for Neural Circuits and Behaviour, London, United Kingdom; [§]Department of Physiology, Anatomy and Genetics, University of Oxford, Oxford, United Kingdom

**Competing interests:** The authors declare that no competing interests exist.

## Introduction

How does activity in neural circuits give rise to behaviour? While mammalian brains are composed of many millions (*Herculano-Houzel et al., 2006*) or billions (*Herculano-Houzel, 2009*; *Herculano-Houzel et al., 2007*) of neurons it has long been postulated that they operate – encoding information and controlling behaviour – through the activity of small subsets of those neurons (*Barlow, 1972*). Indeed, the hypothesis that brains use sparse, distributed activity patterns is supported computationally (*Kanerva, 1993*; *Olshausen and Field, 1996*), energetically (*Attwell and Laughlin, 2001*; *Lennie, 2003*; *Schölvinck et al., 2008*), and experimentally (*Barth and Poulet, 2012*; *Olshausen and Field, 2004*; *Wolfe et al., 2010*). A major factor thought to govern such sparse coding is neuronal inhibition (*Haider and McCormick, 2009*; *Isaacson and Scanziani, 2011*) which serves to balance and control recurrent excitation (*Denève and Machens, 2016*; *Haider et al., 2013*; *Murphy and Miller, 2009*; *Packer and Yuste, 2011*; *Pehlevan and Sompolinsky, 2014*; *Sadeh and Clopath, 2020*; *Tsodyks et al., 1997*; *van Vreeswijk and Sompolinsky, 1996*; *Wehr and Zador, 2003*; *Wolf et al., 2014*) and shape neuronal output (*Borg-Graham et al., 1998*; *Cardin et al., 2010*; *Isaacson and Scanziani, 2011*; *Lee et al., 2012*; *Wilson et al., 2012*). Two key questions are therefore: (1) What is the lower bound of activity that can be behaviourally salient? and (2) How does such activity interact with the local network?

Classical microstimulation experiments have demonstrated that focal activation of cortical regions can influence decision-making, providing a direct causal link between neural activity and behaviour (*Cohen and Newsome, 2004*; *Murasugi et al., 1993*; *Salzman et al., 1990*; *Salzman et al., 1992*). This landmark work has been complemented by more recent studies showing that optogenetic activation of dozens to hundreds of cortical neurons can be directly detected (*Huber et al., 2008*; *Histed and Maunsell, 2014*). A further refinement of this approach was provided by patch-clamp recording, which revealed that strong electrical stimulation of even a single neuron can be detected

in a cell-type and spike timing-dependent manner (*Houweling and Brecht, 2008*; *Doron et al., 2014*), to the extent that they can modulate behaviour in sensory-guided tasks (*Tanke et al., 2018*). While these studies have provided important approximations of the numbers of neurons required to trigger and manipulate behaviour, they suffered from several important limitations. Firstly, electrical stimulation techniques are ill-suited to titrating the number of neurons stimulated and offer limited targeting specificity, either indiscriminately activating swathes of cortex or activating single neurons. Secondly, one-photon optogenetic approaches, while limiting direct excitation to genetically defined neurons, only offer post-hoc estimation of the number of neurons stimulated from histology (*Huber et al., 2008*). Finally, in none of these studies has it been possible to carefully assess the impact of the stimulation on the local network, an essential step if we are to understand the link between activity generated by stimulation and behaviour.

A parallel body of work has investigated the influence of neural activity on local networks in vivo, demonstrating that small numbers of active neurons can have a large impact on local network dynamics and brain state. Strong stimulation of single pyramidal neurons in L2/3 has been shown to recruit ~2% of local excitatory neurons and ~30% of local inhibitory neurons (*Kwan and Dan, 2012*). A single pyramidal neuron spike in L5 is estimated to recruit ~28 post-synaptic neurons (*London et al., 2010*) and in L2/3 can trigger strong disynaptic inhibition (*Jouhanneau et al., 2018*). Single neurons can also trigger global switches in brain state (*Li et al., 2009*), influence network synchronisation (*Bonifazi et al., 2009*) and have a direct impact on motor output (*Brecht et al., 2004*). However, work investigating the impact of sparse activation on the local network in vivo has largely been done under anaesthesia (*London et al., 2010*; *Kwan and Dan, 2012*; *Jouhanneau et al., 2018*), which influences state-dependent cortical processing (*Niell and Stryker, 2010*; *Crochet et al., 2011*; *Harris and Thiele, 2011*) and prevents the study of behaviour.

Combining simultaneous targeted stimulation with readout of effects on the local network during behaviour will allow us to define the local network input-output function. This will yield better understanding of neural network operation, analogously to how measuring single-neuron input-output functions has transformed our understanding of information processing in single neurons (*Magee, 2000*; *Poirazi et al., 2003*; *London et al., 2010*; *Major et al., 2013*). Moreover, it will allow us to determine how this network input-output function in turn influences the psychometric sensitivity to neural activity, which theoretical work predicts is crucial for understanding the link between neural circuit activity and behaviour (*Bernardi et al., 2020*; *Bernardi and Lindner, 2017*; *Bernardi and Lindner, 2019*; *Cai et al., 2020*). While some studies combining readout with manipulation have made significant progress in this direction (*Ceballo et al., 2019a*; *Ceballo et al., 2019b*; *Salzman et al., 1990*; *Znamenskiy and Zador, 2013*), they have lacked spatial resolution and targeting flexibility either on the level of readout or stimulation. Measuring network input-output functions at cellular resolution during perception is likely to yield pivotal insights into how neural populations generate behaviour.

Here, we have activated ensembles of varying numbers of neurons in L2/3 barrel cortex of awake mice trained to detect direct cortical photostimulation. We took advantage of recently developed all-optical approaches combining two-photon calcium imaging and two-photon optogenetics with digital holography to allow us to activate specifically targeted ensembles of neurons while simultaneously recording the response of the local network (*Carrillo-Reid et al., 2019*; *Carrillo-Reid et al., 2016*; *Emiliani et al., 2015*; *Mardinly et al., 2018*; *Marshel et al., 2019*; *Packer et al., 2015*; *Russell et al., 2019*; *Shemesh et al., 2017*). We combined this all-optical approach with an operant conditioning paradigm in which animals were required to report the targeted two-photon optogenetic activation of arbitrary ensembles of pyramidal neurons in L2/3 barrel cortex to gain rewards. In trained animals we investigated how behaviour and network response vary as a function of the ensemble size stimulated. We show that animals are sensitive to the activation of surprisingly small numbers of neurons (~14) and demonstrate that activating roughly double this number of neurons (~37) is sufficient for detection to saturate. Moreover, we show that this perceptual threshold is plastic, and decreases with learning. We also demonstrate that while detection rates increase with increasing stimulation, the surrounding network responds with matched suppression which maintains the balance of activation and suppression at a level consistent with spontaneous epochs.

## Results

### Targeted two-photon optogenetic activation of neural ensembles in L2/3 barrel cortex can drive behaviour

To investigate both the behavioural and network response to precisely controlled levels of cortical excitation, we combined an upgraded version of our previously reported all-optical setup (incorporating 3D volumetric imaging using an ETL, a more powerful two-photon (2P) photostimulation laser and an additional light-path for one-photon (1P) photostimulation; see Materials and methods, *Packer et al., 2015*) with an operant conditioning paradigm whereby mice are trained to report the activation of excitatory neurons in barrel cortex with either 1P or 2P optogenetic stimulation (*Figure 1*). We expressed the calcium indicator GCaMP6s and the two-photon activatable somatically-restricted opsin C1V1 in neurons of L2/3 barrel cortex (*Figure 1a*: inset right, *Figure 1—figure supplement 1*, see Materials and methods). This strategy allows us to flexibly activate specific ensembles of neurons in a given cortical population (*Figure 1a*: Pixel STA, *Figure 1—figure supplement 2*) with high spatial resolution (*Figure 1—figure supplement 2a*: HWHM 5 µm laterally, 20 µm axially) while performing simultaneous two-photon calcium imaging. Inspired by previous work (*Histed and Maunsell, 2014*; *Huber et al., 2008*) we devised a training paradigm in which animals were conditioned to detect bulk activation of barrel cortex via 1P photostimulation. After task acquisition, we progressively lowered the 1P stimulation intensity (which reduces the number, reliability and spatial extent of activated neurons (see Materials and methods for photostimulation details)), before transitioning animals to detect 2P photostimulation of specific groups of neurons (*Figure 1b*, *Figure 1—figure supplement 3*). We used a simple un-cued go/catch trial design where pseudo-randomly interleaved go trials (1P or 2P photostimulation) or catch trials (no photostimulation) were delivered after animals successfully withheld licking for a variable period (*Figure 1c*: top, see Materials and methods). On go trials, the presence/absence of licks in the post-stimulus response window was scored as hits/misses, with hits triggering delivery of a sucrose reward (*Figure 1c*: bottom). On catch trials, the presence/absence of licks were scored as false alarms/correct rejects and neither outcome was punished or rewarded.

Animals underwent three training phases (*Figure 1c*: bottom, *Figure 1—figure supplement 3a*), beginning with Phase 1 where 1P go trials and catch trials were interleaved in equal proportions. In the first training session (*Figure 1—figure supplement 3b*), go trials of 10 mW 1P photostimuli were delivered and were automatically rewarded irrespective of the animal's response (*Figure 1—figure supplement 3b*, blue line). Animals readily learned to detect photostimulation as shown by an increase in proactive lick responses post-stimulus but before the automatic reward (delivered at 0.5 s post-stimulus onset) and decreases in reaction time mean and s.d. across the first 20–100 trials (*Figure 1—figure supplement 3c–e*), often within the first session (*Figure 1—figure supplement 3b*). Once animals showed evidence of learning, automatic rewards were turned off and the LED power was sequentially reduced across several daily training sessions from 10 to 0.25 mW resulting in an inverted U-shaped profile of performance as behaviour improved but stimulation powers dropped (*Figure 1—figure supplement 3f,g*). Subsequently, animals were tested on several 'high power' psychometric curve sessions (*Figure 1—figure supplement 5f* 1.65 ± 1.06 sessions, N = 26 mice, 4 mice skipped this step) where intermediate LED powers (250–50 µW) were pseudorandomly interleaved trial-by-trial (*Figure 1—figure supplement 3h*), finally finishing Phase 1 by undergoing several 'low power' psychometric curve sessions (100–20 µW) (*Figure 1—figure supplement 3i–k*, *Figure 1—figure supplement 5f* 2.12 ± 2.07 sessions, N = 26 mice, 4 mice skipped this step). Animals' response rates decreased with decreasing 1P photostimulation powers (*Figure 1—figure supplement 3i* P(Lick) for 100 µW 0.94 ± 0.11 vs 20 µW 0.51 ± 0.29, p=5.96 × $10^{-5}$ Wilcoxon signed-rank test, N = 22 mice) and their reaction times became slower (*Figure 1—figure supplement 3j* reaction time for 100 µW 0.47 ± 0.16 s vs 20 µW 0.58 ± 0.15 s, p=4.61 × $10^{-5}$ Wilcoxon signed-rank test, N = 22 mice that did this step) and increasingly variable (*Figure 1—figure supplement 3k* reaction time s.d. for 100 µW 0.09 ± 0.04 s vs 20 µW 0.16 ± 0.08 s, p=5.46 × $10^{-4}$ paired t-test, N = 22 mice that did this step), although even the lowest LED powers evoked lick rates significantly higher than catch trials (*Figure 1—figure supplement 3i* P(Lick) for catch trials 0.12 ± 0.08 vs go trials of 20 µW 0.51 ± 0.29, p=2.31 × $10^{-4}$, 40 µW 0.77 ± 0.20, p=2.01 × $10^{-4}$, 60 µW 0.90 ± 0.11, p=2.01 × $10^{-4}$, 80 µW 0.94 ± 0.07, p=2.01 × $10^{-4}$, 100 µW 0.94 ± 0.11, p=2.01 × $10^{-4}$, all Wilcoxon signed-

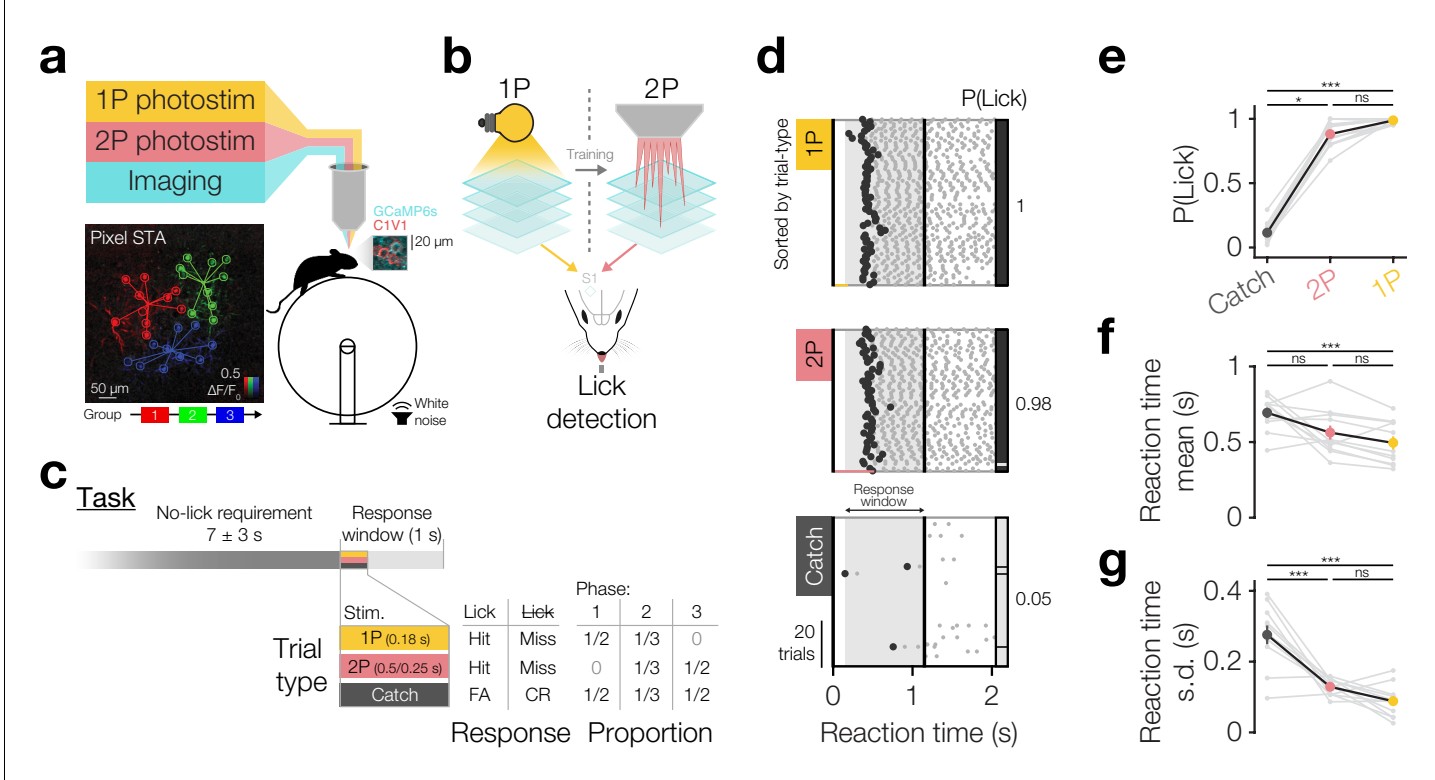

**Figure 1.** Driving behaviour with two-photon optogenetics targeted to ensembles of neurons in L2/3 barrel cortex. (**a**) Schematic of all-optical setup. *Bottom left*: Example of flexible ensemble photostimulation. Three 10 neuron groups in barrel cortex (red, green, blue circles joined by group centroids) were photostimulated sequentially (sequence below Pixel STA). Pixel STA is maximum intensity projection across photostimulus groups of activity in post-photostimulation epoch averaged across trials (N = 3 photostimulus groups, 10 trials each). *Inset right*: sub-region of a full imaging FOV in L2/3 barrel cortex expressing GCaMP6s/C1V1-mRuby. (**b**) Schematic summarising the strategy used to train animals to respond to two-photon optogenetic (2P) stimulation. Mice are first trained to respond to one-photon optogenetic (1P) stimulation of barrel cortex (S1) by licking at an electronic lickometer. The power of 1P illumination is reduced until they can be transitioned onto 2P stimulation targeted to specific ensembles of barrel cortex neurons. (**c**) Structure of the behavioural task (top) and stimulus probabilities, response type contingencies and training phase structures (bottom). Note that stimulus durations, which vary across stimulus types, are not to scale. FA: false alarm; CR: correct reject. (**d**) Lick raster from an example Phase 2 behavioural training session during which a mouse received 2P stimulation trials (pink: 200 neurons), catch trials (grey: no stimulus) and 1P stimulation trials (amber: 0.05 mW, untargeted). Trials were delivered pseudo-randomly (see Materials and methods) but have been sorted by trial type for display. All licks shown in grey with first lick highlighted in black. Hits/false alarms (black) and misses/correct rejects (grey) are indicated as the vertical bar on the right-hand side. Stimulus durations indicated as coloured bars below lick rasters. Behavioural response window indicated as grey shading, label and arrows. (**e– g**), Response rate and reaction time mean and standard deviation for different trial types in final Phase 2 session. Animals detect 1P photostimulation and 2P stimulation targeted to 200 neurons to similar extents, at a level far above chance (catch trials), with similar reaction time mean and standard deviation. (N = 12 mice, 1 session each). Note only animals which responded on >2 catch trials are included for reaction time panels (**f** and **g**) (N = 11 mice, 1 session each). All error bars are s.e.m.

The online version of this article includes the following figure supplement(s) for figure 1:

**Figure supplement 1.** Indicator and opsin expression overlap analysis.

**Figure supplement 2.** Spatial resolution of spiral-scanned two-photon optogenetic activation of Kv2.1-C1V1.

**Figure supplement 3.** 1P training protocol.

**Figure supplement 4.** Example pre-training selection and mapping of 200 neurons in L2/3 barrel cortex.

**Figure supplement 5.** Rapid transfer learning from 1P to 2P optogenetic stimuli.

**Figure supplement 6.** Comparison of behavioural response to somatic and non-somatic C1V1.

rank tests with Bonferroni correction for multiple comparisons, N = 22 mice, average over 1–3 sessions).

At this point, animals were transitioned to Phase 2 where 1P go trials (50 µW), 2P go trials (targeted to 200 and subsequently 100 neurons) and catch trials were pseudorandomly interleaved in equal proportions (*Figure 1c,d*). Before each training session, we selected 200 neurons based on C1V1-mRuby expression in clearly expressing FOVs in superficial L2/3 barrel cortex (~130–230 µm

below pia) and recorded their sensitivity to photostimulation (*Figure 1—figure supplement 4*). For each animal, we selected similarly positioned FOVs across days, but did not specifically target the same neurons (which is challenging due to angular inconsistencies in FOV position across days, see Materials and methods). Using these stimulus patterns for 2P go trials, we trained animals on Phase 2, and subsequently some animals on Phase 3 (only 2P stim and catch trials), for several sessions (*Figure 1d* single session, *Figure 1—figure supplement 5a,b,f* 3.40 ± 1.51 Phase 2/3 sessions 2P 200/100 neurons, N = 12 mice). During this period, we also began interleaving 2P trials stimulating 100 neurons from the 200 neuron group when animals reliably detected 200 neuron stimulations (typically within a single session: 1.08 ± 0.29 sessions to d-prime >1 on 2P 200 neuron trials, N = 12 mice) (*Figure 1—figure supplement 5a,b* performance over time). We found that animals reliably detected 2P photostimulation of both 200 and 100 neurons consistently across time, from the first session (*Figure 1—figure supplement 5c* first 200 neuron d-prime: 2.39 ± 0.86 vs 1, p=1.70 × $10^{-4}$ paired t-test; first 100 neuron d-prime: 2.13 ± 0.93 vs 1, p=1.4 × $10^{-3}$ paired t-test, N = 12 mice) to the last session (*Figure 1—figure supplement 5c* last 200 neuron d-prime: 2.83 ± 0.62 vs 1, p=5.95 × $10^{-7}$ paired t-test; last 100 neuron d-prime: 2.55 ± 0.7 vs 1, p=1.00 × $10^{-5}$ paired t-test, N = 12 mice) and only showed a modest improvement over time (*Figure 1—figure supplement 5c* 200 neuron first 2.39 ± 0.86 vs last 2.83 ± 0.62, p=0.07 paired t-test, N = 12 mice; 100 neuron first 2.02 ± 1.28 vs last 2.86 ± 0.8, p=0.06 Wilcoxon signed-rank test, N = 6 mice with multiple 100 neuron sessions). On their final Phase 2 session (session 2.75 ± 1.42, N = 12 mice) animals detected 2P photostimulation of 200 neurons with high response rates that were similar to 1P photostimulation (*Figure 1d* single session, *Figure 1e* group average, p=1.82 × $10^{-5}$ Friedman test, P(Lick) for 1P 0.99 ± 0.02 vs 2P 0.88 ± 0.10, p=0.19, Bonferroni correction for multiple comparisons, N = 12 mice, 1 session each) and with similar reaction time mean (*Figure 1f* p=6.70 × $10^{-3}$ one-way repeated measures ANOVA, 1P 0.49 ± 0.14 s vs 2P 0.56 ± 0.15 s, p=0.77, Bonferroni correction for multiple comparisons, N = 11 mice, 1 session each, only mice with >2 catch trial responses included) and standard deviation (*Figure 1g* p=4.99 × $10^{-8}$ one-way repeated measures ANOVA, 1P 0.09 ± 0.05 s vs 2P 0.13 ± 0.02 s, p=0.25, Bonferroni correction for multiple comparisons, N = 11 mice, 1 session each, only mice with >2 catch trial responses included). Both 2P and 1P photostimulation evoked higher lick rates than catch trials (*Figure 1e* p=1.82 × $10^{-5}$ Friedman test, P(Lick) for catch 0.11 ± 0.07 vs 2P 0.88 ± 0.10, p=1.61 × $10^{-2}$, vs 1P 0.99 ± 0.02, p=1.04 × $10^{-5}$, Bonferroni correction for multiple comparisons, N = 12 mice, 1 session each) with less variable reaction times (*Figure 1g* reaction time s.d.; p=4.99 × $10^{-8}$ one-way repeated measures ANOVA, catch 0.28 ± 0.09 s vs 2P 0.13 ± 0.02 s, p=5.99 × $10^{-6}$, vs 1P 0.9 ± 0.05 s, p=7.24 × $10^{-8}$, Bonferroni correction for multiple comparisons, N = 11 mice, 1 session each, only mice with >2 catch trial responses included), though only 1P trials showed quicker reaction times (*Figure 1f* p=6.70 × $10^{-3}$ one-way repeated measures ANOVA, catch 0.69 ± 0.11 s vs 1P 0.49 ± 0.14 s, p=5.91 × $10^{-3}$, vs 2P 0.56 ± 0.15 s, p=0.10, Bonferroni correction for multiple comparisons, N = 11 mice, 1 session each, only mice with >2 catch trial responses included). We also note that we found similar response rates in animals expressing non-somatically-restricted C1V1 (*Figure 1—figure supplement 6a* catch subtracted P(Lick) for C1V1-Kv2.1 0.77 ± 0.11 vs C1V1 0.79 ± 0.14, p=0.61 two-sample t-test, N = 12 C1V1-Kv2.1 and 19 C1V1 injected mice), although reaction times were significantly slower for somatically restricted C1V1 (*Figure 1—figure supplement 6b* C1V1-Kv2.1 0.55 ± 0.15 s vs C1V1 0.41 ± 0.08 s, p=2.09 × $10^{-3}$ two-sample t-test, N = 12 C1V1-Kv2.1 and 19 C1V1 injected mice).

Thus, we have demonstrated that two-photon optogenetic stimulation targeted to small ensembles of cortical neurons can reliably drive behaviour and provides a powerful tool for investigating the perceptual salience of different patterns of neural activity.

## Very few cortical neurons are sufficient to drive behaviour

We next tested behavioural sensitivity to the activity of neural ensembles of varying size. To test this we transitioned animals to Phase 3 sessions (2P and catch trials only; *Figure 1c*: bottom) where we precisely titrated the level of activation by targeting different numbers of neurons on a trial-by-trial basis. We again selected 200 neurons on the basis of C1V1-mRuby expression and sub-divided this group into smaller subsets of 100, 75, 50, 25, 10 and 5 neurons (*Figure 2a*). Animals then underwent 2P photostimulation psychometric curve sessions during which, in addition to stimulating the group of 200 neurons, we also pseudorandomly interleaved stimulations of the smaller subsets of neurons (*Figure 2b,c*). We leveraged our ability to simultaneously read out neural activity with calcium

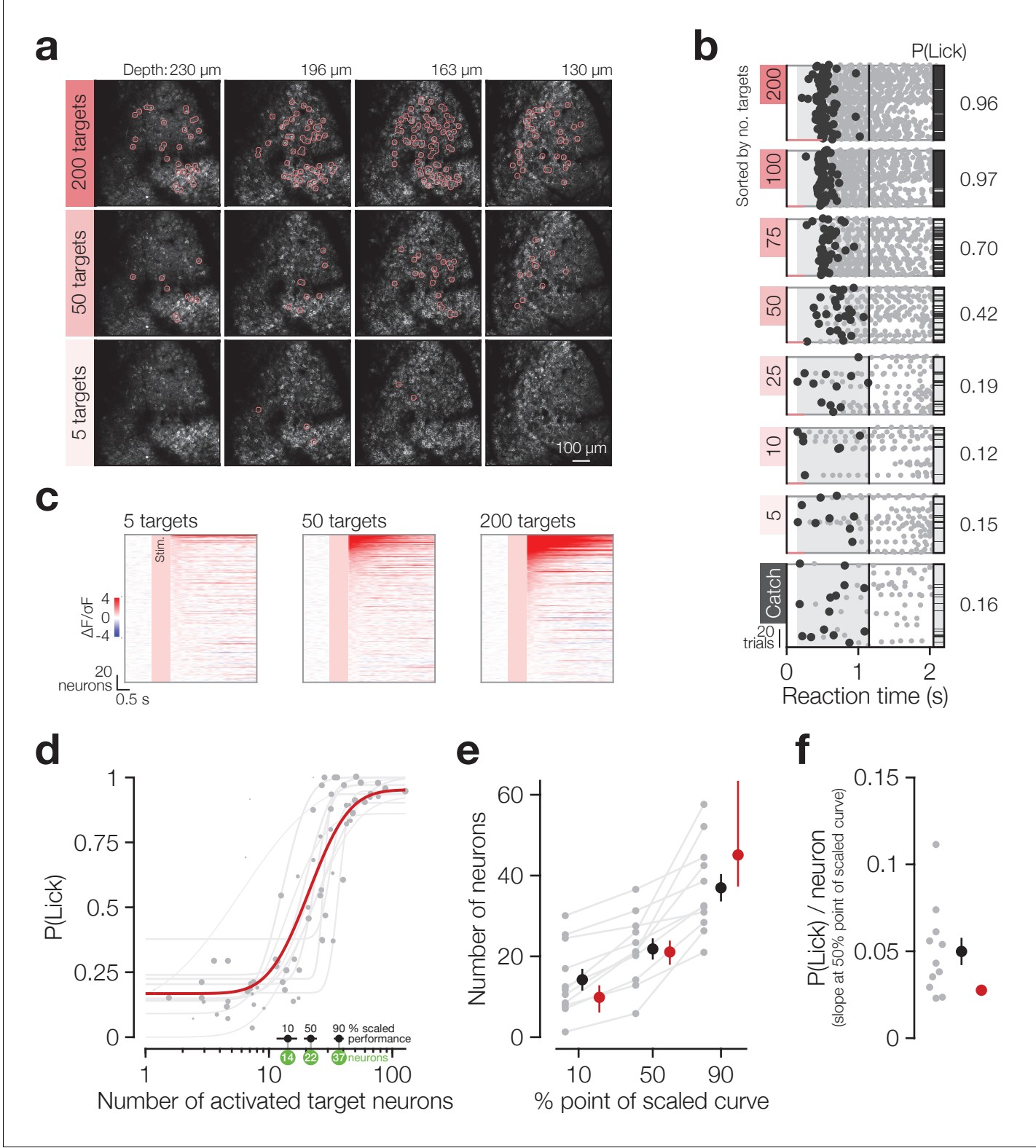

**Figure 2.** Animals detect the targeted activation of tens of neurons. (**a**) Example imaging volumes from an experiment showing 200 (top), 50 (middle) and 5 (bottom) targeted C1V1-expressing neurons. (**b**) Example lick raster concatenating an animal's two psychometric curve testing sessions. Trials were delivered pseudo-randomly (see Materials and methods) but have been sorted by trial type for display. Stimulus durations are indicated by coloured bars along the bottom of each raster. Animals respond on more trials and with less variable timing as more neurons are targeted. (**c**) Example responses across the top 200 most responsive neurons in the 200 target zones (see Materials and methods; *Figure 2—figure supplement 1*). Neurons

*Figure 2 continued on next page*

*Figure 2 continued*

have been sorted separately in each plot. Pink boxes indicate the stimulus artefact exclusion epoch which is consistent across all trial types (see Materials and methods for definition). (**d**) The psychometric function relating the number of activated target neurons to the behavioural detection rate for all 2P psychometric curve sessions. Individual data (grey dots) are grouped by trial type within session (number of target zones) and plotted as the average number of target neurons activated across all trials of each type. Data point size indicates the number of trials of each type (29 ± 8 trials, range 11–44, across data points). Individual psychometric curve fits for each session are plotted (grey lines) weighted by the total number of stimulus trials in the session (202 ± 50 trials, range 97–245, across sessions). The number of neurons required to reach the 10%, 50% and 90% points of these individual scaled psychometric curves are shown as black error bars and green circles about the *x*-axis. The aggregate psychometric curve fit across all trial types, all sessions, is plotted in red. Note that individual curves are often steeper than the aggregate curve. (**e**) The number of neurons required to reach the 10%, 50%, and 90% points of the scaled psychometric curves in (**d**). Grey data points/lines are quantified from individual psychometric curve fits (grey lines in **d**) and summarised by the black error bars. Red data points are quantified from the aggregate psychometric curve fit (red line in **d**) ± confidence intervals. (**f**) The slope at the 50% point of the scaled curves corresponding to the additional probability of detection (P(Lick)) added per target neuron activated. Grey data points are quantified from individual psychometric curve fits (grey lines in **d**) and are summarised by the black error bar. The red circle is quantified from the aggregate psychometric curve fit (red line in **d**) for which no confidence intervals can be calculated (see Materials and methods). *N* = 11 sessions, 6 mice, 1–2 sessions each. All data error bars are mean ± s.e.m. and all fit parameter error bars are estimate ± confidence intervals.

The online version of this article includes the following figure supplement(s) for figure 2:

**Figure supplement 1.** Quantification of neuronal responses.

**Figure supplement 2.** Reaction time standard deviation, but not mean, scales with the number of target neurons activated.

**Figure supplement 3.** Detection of small ensembles of neurons improves across days irrespective of whether the same neurons were targeted.

imaging to refine our estimate of the number of stimulated neurons according to the number of target neurons that were activated averaged across trials. To quantify activation (and suppression; see later sections), we defined thresholds for each neuron on the basis of their response distribution on correct reject catch trials (*Figure 2—figure supplement 1b–g*, see Materials and methods), where no stimulus or lick occurred, and to differentiate between target and background neurons we defined 3D target zones around each 2P photostimulation target co-ordinate (*Figure 2—figure supplement 1a,h–j*, see Materials and methods). This resulted in numbers of activated target neurons that were 0.46 ± 0.20 times that of the number of target zones (averaged across trial types) and decreased with decreasing number of zones as intended (*Figure 2—figure supplement 1i*; see Materials and methods).

Animals' response rates increased sigmoidally with increasing numbers of target neurons activated (*Figure 2b* single animal, *Figure 2d* all sessions) and both individual sessions and aggregate data across sessions were well fit by log-normal sigmoid psychometric functions (*Figure 2d* grey dots/lines: individual data/fits, $R^2$ = 0.91 ± 0.09, *N* = 11 sessions, 6 mice, 1–2 sessions each; red line: aggregate fit, $R^2$ = 0.72, cross-validated $R^2$ = 0.67 ± 0.23, *N* = 10,000 permutations, see Materials and methods for fit details). Using these fitted psychometric functions, we estimate that animals can detect the activation of a minimum of ~14 neurons at their perceptual threshold (*Figure 2d,e* 10% point of curve: individual fits 14.2 ± 8.96 neurons, *N* = 11 sessions, 6 mice, 1–2 sessions each; aggregate fit 9.86 [95% CI: 6.09 12.8] neurons), with only roughly double this number of neurons (~37) required to saturate performance (*Figure 2d,e* 90% point of curve: individual fits 37.0 ± 11.3 neurons, *N* = 11 sessions, 6 mice, 1–2 sessions each; aggregate fit 45.1 [95% CI: 37.3 63.5] neurons). At the psychometric function's 50% point (*Figure 2d,e* 50% point of curve: individual fits 21.8 ± 8.71 neurons, *N* = 11 sessions, 6 mice, 1–2 sessions each; aggregate fit 21.1 [95% CI: 17.9 23.9] neurons) this results in a very steep slope, with ~0.05 probability of licking added per neuron stimulated (*Figure 2f* individual fits 0.05 ± 0.03; aggregate fit 0.03). This is notably steeper for individual fits than the aggregate fit (0.05 ± 0.03 vs 0.03, p=9.77 × $10^{-3}$ Wilcoxon signed rank test, *N* = 11 sessions, 6 mice, 1–2 sessions each). Mean reaction times did not vary as fewer neurons were activated (*Figure 2—figure supplement 2a* β = −0.02, $R^2$ = 0.02, p=0.24), although they did become more variable (*Figure 2—figure supplement 2b* β = −0.05, $R^2$ = 0.30, p=1.41 × $10^{-6}$). This demonstrates that animals are exquisitely sensitive to the activation of small numbers of cortical neurons and can read out surprisingly small changes in cortical activity levels.

We next addressed the question of how flexible this perceptual threshold is and how specific it is to neurons used for training during preceding sessions. After training animals to detect the activation of hundreds of barrel cortex neurons, we asked whether their ability to detect the activation of

small subsets of these neurons improved across multiple subsequent days, and whether learning was specific to neurons targeted on each day. In a second cohort of animals, we identified and activated the same neurons reliably across multiple days (*Figure 2—figure supplement 3a*) and measured the detection rate across sessions (*Figure 2—figure supplement 3b*), whereas in the first cohort mentioned above we moved FOV for each session and stimulated different neurons. Across all animals there was a consistent improvement in detection rate (*Figure 2—figure supplement 3c* P (Lick) on Session 1: 0.17 ± 0.18 vs Session 2: 0.28 ± 0.25, p=6.29 × $10^{-3}$ paired t-test, $N$ = 14 mice testing the same 30 neurons and 5 mice testing different groups of 25 and 50 neurons) which did not differ depending on whether the same or different neurons were stimulated across sessions (*Figure 2—figure supplement 3d* P(Lick) improvement for same: 0.12 ± 0.18 vs different: 0.09 ± 0.18, p=0.84 Mann Whitney U-Test, $N$ = 14 mice testing the same 30 neurons and 5 mice testing different groups of 25 and 50 neurons).

These experiments use targeted stimulation of cortical neurons to describe the behavioural input-output function for our task, and suggest that the lower bound for detection is a small number of neurons (~14 neurons) and the psychometric function is very steep (saturating at ~37 neurons). We also demonstrate that animals' ability to detect small numbers of neurons improves with training and that this improvement is not limited to targeted neurons.

## Suppression in the local network balances target activation

We took advantage of our ability to simultaneously record activity in both the targeted and untargeted 'background' neurons to investigate how local network activity influences or depends on behavioural performance. Using the same activation thresholds, suppression thresholds and target definitions described earlier (*Figure 2—figure supplement 1*, see Materials and methods) we calculated the proportion of activated and suppressed neurons on each trial averaged across trials of each type (*Figure 3—figure supplement 1a,e*). Splitting trials by hits and misses we found that hits were associated with more activation (*Figure 3—figure supplement 1a,b* P(activated) background on hits 4.94 × $10^{-2}$ ± 8.63 x $10^{-3}$ vs misses 3.83 × $10^{-2}$ ± 5.04 x $10^{-3}$ on 50 target trials, p=1.95 × $10^{-3}$ Wilcoxon signed-rank test, $N$ = 11 sessions, 6 mice, 1–2 sessions each) and less suppression (*Figure 3—figure supplement 1e,f* P(suppressed) background on hits 3.53 × $10^{-2}$ ± 4.84 x $10^{-3}$ vs misses 3.99 × $10^{-2}$ ± 5.5 x $10^{-3}$ on 50 target trials, p=2.23 × $10^{-2}$ paired t-test, $N$ = 11 sessions, 6 mice, 1–2 sessions each) than misses. However, since hits are associated with stereotyped behaviours (licking, whisking, face movements etc.), and significant movement and reward-related activity has been observed in primary sensory cortical areas (*Shuler and Bear, 2006*; *Niell and Stryker, 2010*; *Musall et al., 2019*; *Steinmetz et al., 2019*; *Stringer et al., 2019*; *Zatka-Haas et al., 2020*), we reasoned that such differences might be accounted for by the behaviours themselves irrespective of our manipulations. Indeed we found that in background neurons the level of activation recruited on hits post-photostimulation was not different from false alarms on catch trials where animals licked but no neurons were photostimulated (*Figure 3—figure supplement 1d* P(activated) background on 50 target hit 4.92 × $10^{-2}$ ± 9.05 x $10^{-3}$ vs catch false alarm 4.83 × $10^{-2}$ ± 1.04 x $10^{-2}$, p=0.52 paired t-test, $N$ = 10 sessions, 6 mice, 1–2 sessions each, 1 session without any catch false alarms excluded), irrespective of how many neurons we activated (*Figure 3—figure supplement 1a*). The amount of suppression also did not differ between hits post-photostimulation and false alarms on catch trials (*Figure 3—figure supplement 1h* 50 neuron hit vs catch false alarm: 3.51 × $10^{-2}$ ± 5.05 x $10^{-3}$ vs 3.37 × $10^{-2}$ ± 6.99 x $10^{-3}$, p=0.27 paired t-test, $N$ = 10 sessions, 6 mice, 1–2 sessions each, one session without any catch false alarms excluded), although there was some modulation of this difference with the number of neurons activated (*Figure 3—figure supplement 1e*). It therefore seemed possible that a significant amount of the stimulus-evoked activity that we read out in background neurons was influenced by lick-related behaviours. In line with this, we found that a large fraction of neurons showed activity which was modulated by spontaneous licking (*Figure 3—figure supplement 1i–k* 46% ± 11 of neurons lick modulated, $N$ = 11 sessions, 6 mice, 1–2 sessions each, see Materials and methods) with neurons showing both positive correlation (*Figure 3—figure supplement 1j,l,m* 2.83 × $10^{-2}$ ± 1.81 x $10^{-2}$ lick correlation for all positively lick modulated neurons, $N$ = 9547 neurons) and negative correlation (–2.52 x $10^{-2}$ ± 1.51 x $10^{-2}$ lick correlation for all negatively modulated neurons, $N$ = 4365 neurons).

Unfortunately, the temporal resolution of calcium imaging does not allow us to tease apart the direction of causality in our data, that is whether this activity causes, or is caused by motor output.

However, given the literature demonstrating behavioural output-related activity in sensory cortices (*Musall et al., 2019*; *Steinmetz et al., 2019*; *Stringer et al., 2019*), and the fact that manipulating this activity has no effect on behavioural choices (*Zatka-Haas et al., 2020*), we were concerned that a significant amount of network activity might result from movement rather than causing it. This would be problematic for our interpretation since the amount of lick contamination will vary by trial type (number of neurons activated) in a manner that correlates with the variable under study (due to the increased P(Lick) with number of neurons activated *Figure 2d*). To take account of this, we devised a hit:miss matching procedure which removes the variance in hit:miss ratio across trial types by ensuring that all trial types have a 50:50 ratio of hits:misses (*Figure 3—figure supplement 1o*, see Materials and methods). This is achieved for trials of a given type (i.e. a low P(Lick) trial type: 10 activated neurons) by matching the number of trials of the minority response type (i.e. hits) with random resamples, of the same number, of majority response-type trials (i.e. misses) and averaging network response metrics across resamples. Following this procedure should ensure that all trial types have the same proportion of data contaminated by lick responses and any variation in network response across trial types remaining should be due to the variation in number of target neurons activated.

Using this procedure, we investigated how the network response varies as a function of stimulated ensemble size beyond its stereotyped modulation by the behavioural response. Taking the network as a whole (including target neurons), we found that photostimulation causes both activation (*Figure 3a* right inset; P(activated) all neurons on photostimulus $5.91 \times 10^{-2} \pm 1.20 \times 10^{-2}$ vs catch $4.82 \times 10^{-2} \pm 6.65 \times 10^{-3}$ trials averaged across all trial types, p=$1.17 \times 10^{-3}$ paired t-test, $N = 10$ sessions, 6 mice, 1–2 sessions each) and suppression (*Figure 3b* right inset; P(suppressed) all neurons on photostimulus $4.58 \times 10^{-2} \pm 4.90 \times 10^{-3}$ vs catch $3.97 \times 10^{-2} \pm 4.80 \times 10^{-3}$ trials averaged across all trial types, p=$1.41 \times 10^{-4}$ paired t-test, $N = 10$ sessions, 6 mice, 1–2 sessions each) and that both scale with the number of neurons activated (*Figure 3a,b* P(activated) all neurons: $\beta = 7.54 \times 10^{-3}$, $R^2 = 0.24$, p=$2.36 \times 10^{-5}$; P(suppressed) all neurons: $\beta = 3.00 \times 10^{-3}$, $R^2 = 0.25$, p=$1.54 \times 10^{-5}$, $N = 10$ sessions, 6 mice, 1–2 sessions each). When we analysed only background neurons (i.e. excluding targets from the calculation), we found that while photostimulation does cause both activation and suppression in the background network (*Figure 3d,e* right insets; P(activated) network on photostimulus $4.40 \times 10^{-2} \pm 6.89 \times 10^{-3}$ vs catch $4.02 \times 10^{-2} \pm 5.60 \times 10^{-3}$ trials, p=$2.31 \times 10^{-3}$ paired t-test; P(suppressed) background on photostimulation $3.81 \times 10^{-2} \pm 4.60 \times 10^{-3}$ vs catch $3.27 \times 10^{-2} \pm 4.60 \times 10^{-3}$ trials, p=$1.06 \times 10^{-4}$ paired t-test, all averaged across trial types, $N = 10$ sessions, 6 mice, 1–2 sessions each), only background network suppression scales with the number of activated target neurons (*Figure 3e* $\beta = 2.95 \times 10^{-3}$, $R^2 = 0.31$, p=$7.79 \times 10^{-7}$), whereas activation does not (*Figure 3d* $\beta = 1.70 \times 10^{-3}$, $R^2 = 0.05$, p=0.08). Moreover, activation and suppression have distinct spatial profiles, with significant suppression occurring over a much broader area (*Figure 3—figure supplement 3*). This suggests that the network reacts to suppress the spread of activation triggered by our photostimulation in a graded manner which tracks the activation strength, whereas network activation changes to a smaller extent. Indeed, across all neurons in the population we see that there is a consistent balance of activation and suppression across all target activation levels (*Figure 3c* P(Activated)/P(Suppressed) across all neurons $\beta = 7.23 \times 10^{-2}$, $R^2 = 0.05$, p=0.06) that remains similar to the rates observed spontaneously during catch trials (*Figure 3c* right inset; P(Activated)/P(Suppressed) on photostimulation $1.30 \pm 0.28$ vs catch $1.24 \pm 0.25$ trials across all neurons, p=0.25 paired t-test, $N = 10$ sessions, 6 mice, 1–2 sessions each).

These results suggest that activating target neurons produces suppression in the surrounding network which maintains homeostasis of background activity.

## Behaviour tracks target neuron activity despite constant, matched suppression in the local network

As it has been suggested that both excitation and inhibition of cortical neurons can drive behaviour (*Doron et al., 2014*), and since target activation and network suppression correlate in our dataset (*Figure 3e*), we finally asked which of the factors that we have analysed best correlates with animals' behavioural responses. Using the hit:miss matched data described above, we modelled behavioural response rates as a function of target activation, background activation and background suppression (*Figure 4a–c*), cross-validating across training and test datasets to assess the goodness and generalisability of the fits (see Materials and methods). The strongest and most generalisable predictor of

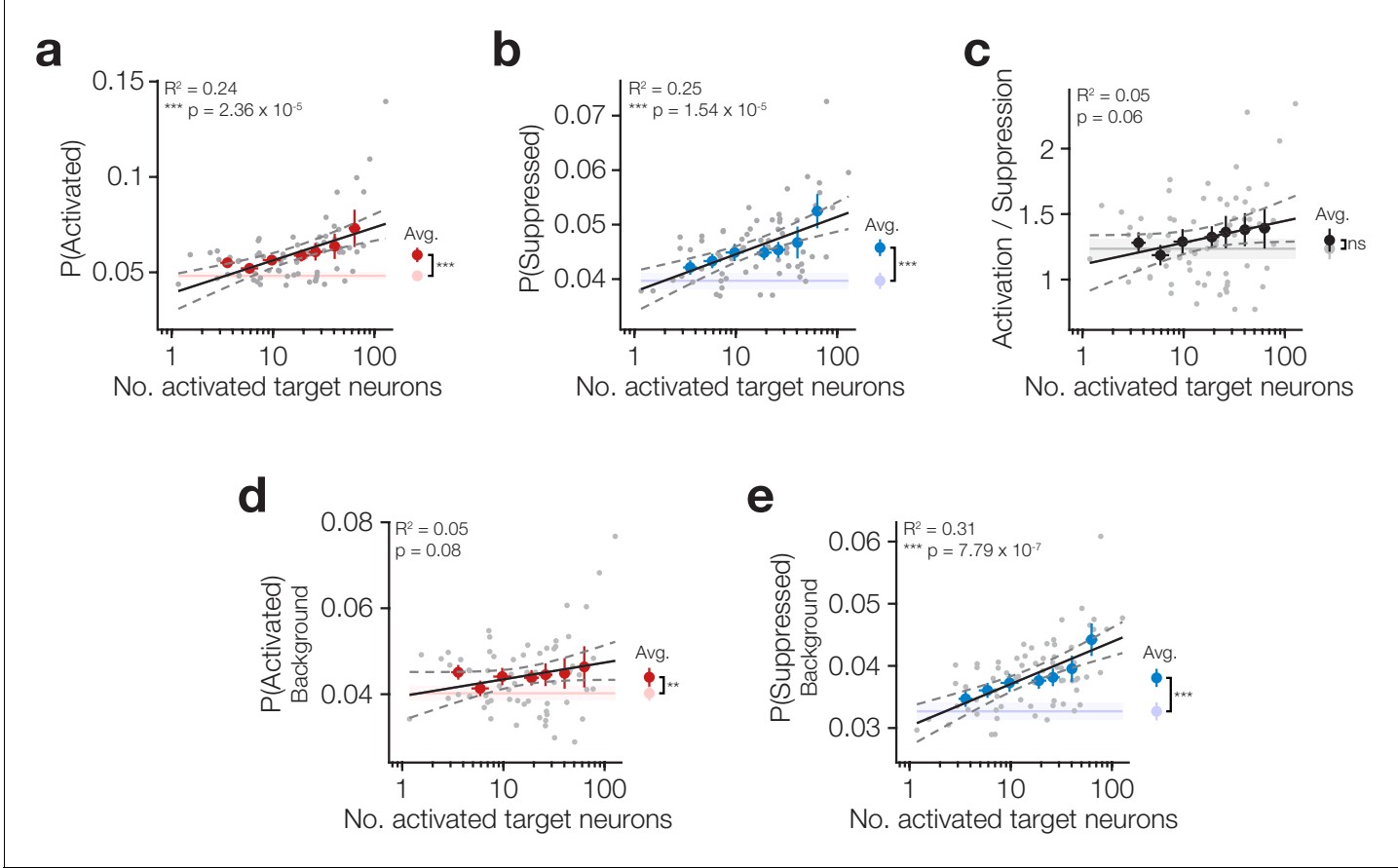

**Figure 3.** Increasing target activation is matched by background network suppression. (a) The proportion of neurons activated across all neurons (targets and background) increases as more target neurons are activated. *Inset right:* average activation across all trial types is increased on stimulus trials compared to catch. (b) The proportion of neurons suppressed across all neurons (targets and background) increases as more target neurons are activated. *Inset right:* average suppression across all trial types is increased on stimulus trials compared to catch. (c) The ratio of activation and suppression is similar to that observed on catch trials (*inset right*) and is not strongly modulated by the number of activated target neurons. (d) Stimulation of target neurons causes mild activation of background neurons (targets excluded; *inset right*) but this is not modulated by the number of target neurons activated. (e) Stimulation of target neurons causes suppression of background neurons (targets excluded; *inset right*) which increases as more target neurons are activated. All data are hit:miss matched to remove potential lick signals (see *Figure 3—figure supplement 1* and Materials and methods). For all plots N = 11 sessions, 6 mice, 1–2 sessions each. Some trial types from some sessions are excluded for having too few hits or misses to be able to match the hit:miss ratio. Error bars and shading are s.e.m; data points, error bars and linear fits are stimulus trials, shading is catch trials; grey data points: individual trial types, individual sessions; coloured error bars: data averaged within trial type (number of target zones) across sessions; linear fits are to individual data points; fits are reported ± 95% confidence intervals.
The online version of this article includes the following figure supplement(s) for figure 3:

**Figure supplement 1.** Comparison of network activity on hits and misses for both threshold go trials and catch trials in an effort to quantify and account for lick responses.
**Figure supplement 2.** Neuropil subtraction has a small effect on response amplitude but it is not the sole cause of negative going responses.
**Figure supplement 3.** Activation and suppression have different spatial profiles.

behavioural responses was target neuron activation, which had significantly positive $R^2$ across both training and test sets during cross-validation (*Figure 4a,d* train $R^2$: 0.71 ± 0.06, p=0 permutation test vs 0; test $R^2$: 0.66 ± 0.23, p=1.68 × $10^{-2}$ permutation test vs 0, N = 10,000 train:test splits). Network suppression had mild predictive power on training data, but none on testing data (*Figure 4c,d* train $R^2$: 0.16 ± 0.06, p=2.50 × $10^{-3}$ permutation test vs 0; test $R^2$: 0.01 ± 0.39, p=0.39 permutation test vs 0, N = 10,000 train:test splits) which might be explained by its correlation with target activation (*Figure 3e*). Network activation was a poor predictor of both training and testing data, suggesting that it had little influence on behavioural performance (*Figure 4b,d* train $R^2$: −0.05 ± 0.05, p=0.11 permutation test vs 0; test $R^2$: −0.21 ± 0.44, p=0.22 permutation test vs 0, N = 10,000 train:test

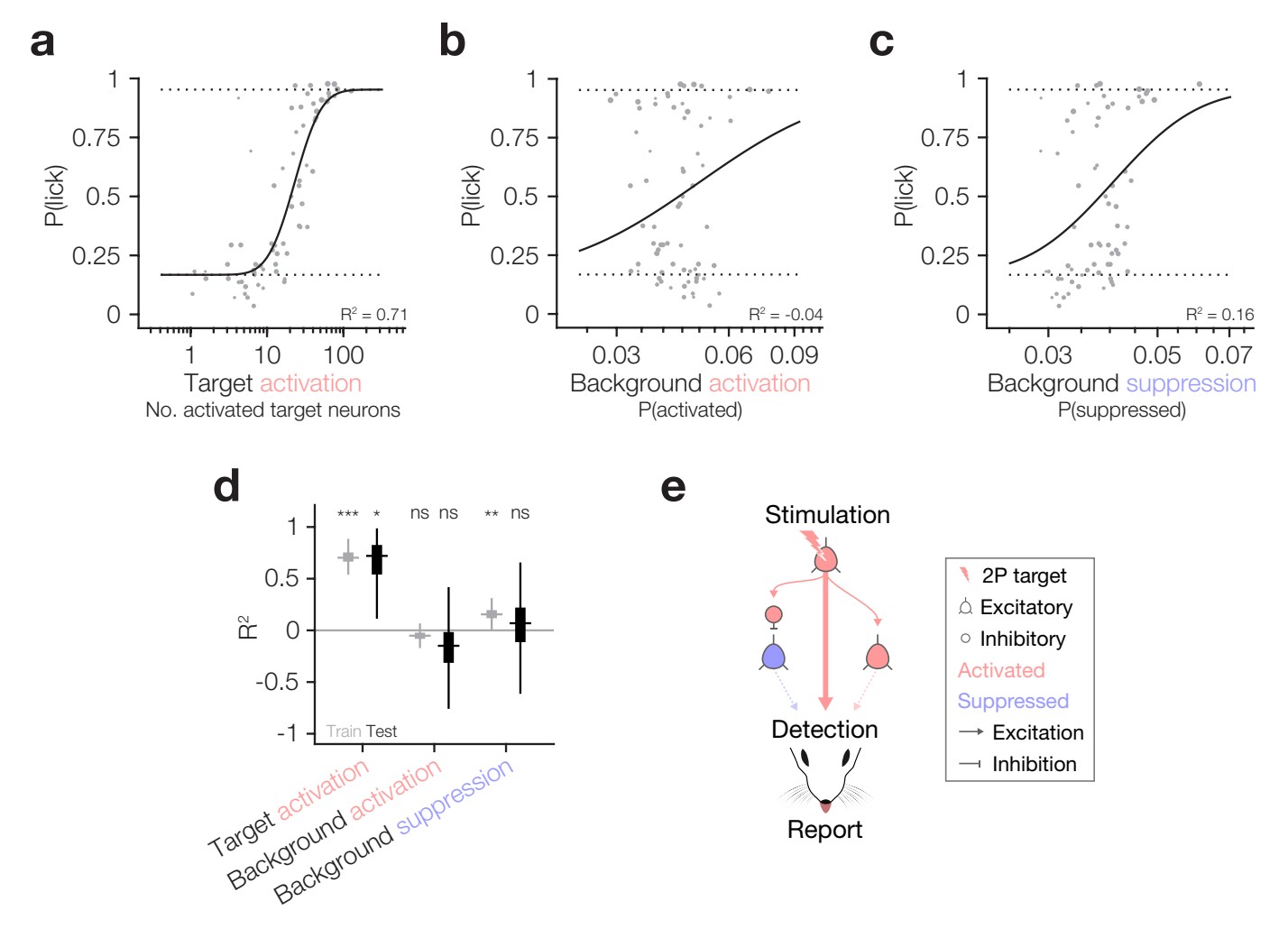

**Figure 4.** Behaviour follows the activity of targeted ensembles despite matched suppression in the local network. (a–c) Psychometric curve fits relating behavioural detection to the number of targets activated (a), the proportion of the background network activated (b) and the proportion of the background network suppressed (c). Solid lines: psychometric curve fit; dotted lines: fixed lapse rate (upper) and false alarm rate (bottom) for psychometric fit. Note that all neural data has been hit:miss matched (see Materials and methods) so the effective P(lick) for all datapoints is 0.5; however, for each datapoint we fit the actual recorded P(lick). The similarity of panel (a), which is hit:miss matched, to *Figure 2d*, which is not, demonstrates that the contribution of lick signals to the relationship between target activation and behaviour is negligible. Fits and $R^2$ values reported are quantified on all data (compared to cross-validated fits in following panels). For these panels, *N* = 11 sessions, 6 mice, 1–2 sessions each. Some trial types from some sessions are excluded for having too few hits or misses to be able to match the hit:miss ratio. (d) Variance explained ($R^2$) by the three predictors in (a–c) during the training (grey) and testing (black) phases of cross-validation (10,000 permutations of 80:20 train:test split). Only target activation strongly and reliably explains behaviour across both training and testing. Background suppression is mildly predictive of behaviour in model training datasets, but this relationship does not generalise to test datasets. Background activation does not explain behaviour. Boxplots are median with 25th and 75th percentile boxes and whiskers extending to the most extreme data points not considered outliers (see Materials and methods). (e) Schematic summarising the three tested routes from cortical activation to behavioural report highlighting that only the activity of target neurons has any reliable influence on behaviour despite matched suppression in the local network.

splits). Thus, within the cortical region that we can observe and manipulate we have tested three of the main pathways by which activity in cortex could influence downstream circuitry to ultimately drive behaviour: (1) output of directly activated target neurons; (2) output of background neurons synaptically activated by target neurons; (3) suppression of the output of background neurons through disynaptic inhibition by interneurons activated by target neurons (*Figure 4e*). Our results indicate that the most robust effect is the number of activated target neurons (1), with only a minor impact of indirectly modulated neurons in the local network.

## Discussion

By precisely titrating the number of activated neurons to be detected, we have demonstrated that the psychometric function for detecting cortical activity is sensitive, with only tens of neurons required to drive behaviour, and it is steep, with only an approximate doubling in number sufficient to drive asymptotic behavioural performance. Simultaneous imaging of the surrounding network has allowed us to show that, despite this exquisite behavioural sensitivity, the dominant network response matches the activation of targeted neurons with local suppression, flattening the network input-output function to maintain the level of network activation within the spontaneous range. These results support the sparse coding hypothesis (*Barlow, 1972*; *Barth and Poulet, 2012*; *Kanerva, 1993*; *Olshausen and Field, 1996*), demonstrate that the local network operates in an inhibition-stabilised regime (*Denève and Machens, 2016*; *Murphy and Miller, 2009*; *Ozeki et al., 2009*; *Pehlevan and Sompolinsky, 2014*; *Sanzeni et al., 2020*; *Tsodyks et al., 1997*; *van Vreeswijk and Sompolinsky, 1996*; *Wolf et al., 2014*), and suggest a high storage capacity for recurrent networks (*Hopfield, 1982*; *Lefort et al., 2009*; *Peron et al., 2020*). This combination of features likely maximises perceptual sensitivity while minimising erroneous detection of background activity, thus avoiding hallucinations (*Carbon, 2014*; *Cassidy et al., 2018*; *Corlett et al., 2019*; *Friston, 2005*), runaway excitation (*Rose and Blakemore, 1974*; *Treiman, 2001*; *Ziburkus et al., 2006*) and reducing cortical energy requirements (*Schölvinck et al., 2008*). We also show that the lower bound of detectable activity is not fixed, but can improve with training in a way that can generalise to other neurons in the surrounding network. This suggests that perceptual learning can reroute cortical resources to meet even the most stringent task demands and generalise to other potentially relevant neurons, increasing cognitive flexibility. This also demonstrates that the brain's ability to 'learn to learn' (*Behrens et al., 2018*; *Harlow, 1949*) can extend even to arbitrary activity patterns.

### Activation of only a small number of neurons is required to reach perceptual threshold

We leveraged our all-optical system to precisely target different numbers of neurons for two-photon optogenetic activation in order to define the minimum number that is sufficient to drive behaviour. The perceptual threshold we measured is remarkably low: mice can detect the activation of only ~14 cortical neurons. This number is substantially lower than the threshold estimated for one-photon activation of layer 2/3 neurons in rodent barrel cortex (~60; *Huber et al., 2008*). This might be explained by the fact that we tailor our optogenetic stimulus for activation of individual neurons, whereas one-photon stimulation diffusely activates an entire population of neurons (many of which will receive subthreshold levels of photocurrent). Moreover, our stimuli drive double the number of action potentials (~10 in each neuron) which Huber et al. show increases detectability, though we note that our minimum threshold of ~140 action potentials (10 action potentials in 14 neurons) is still lower than the ~300 that they report. Furthermore, we produce comparatively more clustered activation of stimulated neurons in our ensembles (confined to a ~500 × 500 x 100 μm volume) compared to the more dispersed neuron locations used in Huber et al. (potentially across the whole of S1), which may result in different recruitment of both local and downstream targets due to changes in intra- and inter-laminar connection probability with distance (*Holmgren et al., 2003*; *Lefort et al., 2009*; *Perin et al., 2011*; *Thomson and Lamy, 2007*; *Yoshimura et al., 2005*). Finally, the *Huber et al., 2008* study relied on post-hoc histological estimates which, as noted by the authors, may have a significant margin of error.

On the other hand, our estimate of the perceptual threshold is an order of magnitude higher than single-cell patch-clamp experiments demonstrating that strong activation of single neurons can in some cases lead to a behavioural report (*Doron et al., 2014*; *Houweling and Brecht, 2008*; *Tanke et al., 2018*). Several key differences could account for this discrepancy. First, these studies mostly stimulated L5 neurons, which serve as the output neurons of the cortical circuit and therefore may drive behaviour more reliably compared to the L2/3 ensembles we target. Indeed, such a dichotomy is confirmed by a recent study comparing the behavioural influence of functionally defined ensembles in L2/3 with those in L5 (*Marshel et al., 2019*). Secondly, these studies report significant variability in the ability of single neurons to drive behaviour, with many neurons having no effect, and with the most potent behavioural effects being limited mainly to fast-spiking putative interneurons (*Doron et al., 2014*). Additional factors which may play a role include differences in

level of stimulation of individual neurons, differences in the training protocol, or species differences between mice and rats.

Interestingly, the reaction times to optogenetic stimuli that we report (0.4–0.7 s) are comparable to, although slightly slower than, those reported for detection of whisker stimuli in mice (0.3–0.4 s; *Chen et al., 2013a*; *Hires et al., 2015*; *O'Connor et al., 2010*; *Sachidhanandam et al., 2013*) suggesting that they may be processed differently. This could be because sensory stimuli more robustly recruit many neurons distributed over several parallel thalamo-cortical pathways which provide input to multiple cortical layers (*Feldmeyer et al., 2013*; *Petersen, 2007*), including direct projections from thalamus to cortical output L5 (*Constantinople and Bruno, 2013*), whereas our optogenetic stimuli will only activate small numbers of neurons in L2/3. Moreover, sensory stimuli will likely drive patterns of activity that respect cortical wiring for transmission of sensory information and so may propagate more potently downstream. Future work comparing reaction times to targeted activation of sensory-evoked or random ensembles of neurons in the same animal will yield insight into any distinction between naturalistic and artificial neural activity in driving behaviour. Our results also reveal reduced reaction time for non-somatically-restricted C1V1 with little difference in response rate. This difference in reaction time may be due to increased off-target activation via processes traversing the photostimulation volume as well as larger photocurrent in target neurons through activation of opsin in neuronal compartments in addition to the soma (dendrites/axon). It is possible that this does not result in an increased response rate because animals' performance is saturated at its upper bound (allowing for a lapse rate that is independent of the salience of stimuli).

What are the functional implications of the perceptual threshold we have defined? The fact that in our study (see also *Daie et al., 2019*; *Gill et al., 2020*; *Marshel et al., 2019*) the lower perceptual bound is well above a single neuron suggests that perceptual thresholds are tuned to minimise conscious perception of the spontaneous neural activity which often co-exists alongside stimulus activity (*Bernander et al., 1991*; *Destexhe and Paré, 1999*; *Destexhe et al., 2003*; *London et al., 2010*; *Musall et al., 2019*; *Shadlen and Newsome, 1994*; *Stringer et al., 2019*; *Tolhurst et al., 1983*; *Waters and Helmchen, 2006*). If the perceptual apparatus is sensitive to single-neuron activation, this may lead to erroneous detection of background cortical activity. While the stimuli in our study do not explicitly mimic sensory evoked activity, such false positives in our behavioural paradigm may be related to the hypothesised role of false sensory percepts in generating hallucinations, which impair normal cognitive function and are associated with an array of pathological conditions (*Chaudhury, 2010*; *Kumar et al., 2009*; *Llorca et al., 2016*). The threshold of ~14 neurons could therefore be important for avoiding perceptual false positives caused by spontaneous background network activity.

The relatively low perceptual threshold may also have significant computational advantages. A large body of theoretical work has suggested that the brain may use a sparse coding scheme to represent information (*Barlow, 1972*; *Kanerva, 1993*; *Olshausen and Field, 1996*). Our demonstration that the perceptual threshold (~14 neurons) is much lower than the dimensionality of barrel cortex (~400,000 neurons in barrel cortex; *Hooks et al., 2011*; *Meyer et al., 2013*; ~2000 neurons in superficial layers of a barrel; *Lefort et al., 2009*), but also significantly higher than a single neuron, is consistent with computational theories proposing that individual items are represented sparsely compared to the dimensionality of the space, but also that they are not represented by single elements of that space (*Baum et al., 1988*; *Kanerva, 1993*; *Olshausen and Field, 2004*; *Palm, 1980*). Our work is also consistent with work suggesting that barrel cortex can use ensembles on this scale to robustly encode sensory information (*Hires et al., 2015*; *Mayrhofer et al., 2015*; *Panzeri et al., 2014*; *Stüttgen and Schwarz, 2008*). Such sensitivity is beneficial as it suggests that recurrent networks like cortical L2/3 have a high-storage capacity (*Hopfield, 1982*; *Lefort et al., 2009*; *Ko et al., 2011*; *Harris and Mrsic-Flogel, 2013*; *Cossell et al., 2015*; *Peron et al., 2020*) allowing the brain to represent many patterns independently (*Amit et al., 1985a*; *Amit et al., 1985b*; *McEliece et al., 1987*; *Brunel, 2016*; *Folli et al., 2016*).

## The network input-output function for perception is steep

By recording the response of the local network while carefully titrating the number of targeted neurons, we have defined the network input-output function for perception in our task. This function is sigmoidal and remarkably steep, saturating at only ~37 neurons. These results echo similarly steep perceptual input-output functions found in other systems (*Gill et al., 2020*; *Marshel et al., 2019*)

but are much steeper than estimated in barrel cortex for one-photon optogenetic stimulation (*Huber et al., 2008*). Again, this discrepancy may be due to a range of factors associated with one-photon photostimulation, from the spatially diffuse (and largely subthreshold) nature of the activation to differences in network cooperativity associated with our more clustered stimulation patterns. Indeed, the highly synergistic activation of a local network of densely interconnected excitatory neurons (*Cossell et al., 2015*; *Douglas et al., 1995*; *Ko et al., 2011*), rapidly followed by suppression in the local network (*London et al., 2010*; *Kwan and Dan, 2012*; *Chettih and Harvey, 2019*) likely mediated by disynaptic inhibition (*Jouhanneau et al., 2018*; *Mateo et al., 2011*; *Silberberg and Markram, 2007*), may be the basis for the steep and saturating input-output function we have observed.

This steep input-output function may have significant functional consequences. While allowing rejection of noise due to spontaneous activity (see the previous section), it also enables perceptual detection of relevant activity with high sensitivity and efficiency, and yet avoids further unnecessary engagement of the network with additional stimulation. This may represent a circuit mechanism for optimising the canonical trade-off in a sensory system subject to noise (*Bialek, 2012*) between minimising false positives (a response when there is no signal: a 'false alarm') and minimising false negatives (missing a signal when there is one present). The steepness of the input-output function and the low number of neurons at saturation are also consistent with optimal energy efficiency (*Attwell and Laughlin, 2001*; *Lennie, 2003*). Indeed, our results offer cellular-resolution support for the proposal that the energy associated with conscious perception is surprisingly low (*Schölvinck et al., 2008*), since our data predict that the number of additional neurons required to allow a subconsciously processed sensory stimulus to be consciously perceived will be low.

While our behavioural paradigm relies on stimulation of artificially defined ensembles of neurons, we maintain that our results add general insight as to how neural activity can underlie flexible behaviour in barrel cortex since: (1) bulk optogenetic activation of pyramidal neurons in sensory areas can readily replace trained sensory stimuli with minimal behavioural impact (*Ceballo et al., 2019b*; *O'Connor et al., 2013*; *Sachidhanandam et al., 2013*), suggesting that the activity evoked is not so alien as to confuse behavioural processing; (2) the order of magnitude of the numbers that we report corresponds closely with the estimated number of barrel cortex neurons required to decode tactile stimuli of various types (*Hires et al., 2015*; *Mayrhofer et al., 2015*; *Stüttgen and Schwarz, 2008*) suggesting that there is limit to this system's sensitivity that can be found through both observation and causal manipulation; (3) irrespective of the exact numbers, the steepness of the psychometric function suggests a fine distinction between whether neural activity is perceptible or not which is indicative of a highly sensitive yet specific sensory system, as has also been shown for optogenetic stimuli mimicking sensory (visual) percepts (*Marshel et al., 2019*); (4) the unique ability of two-photon optogenetics to specifically target the same, or different, ensembles of neurons throughout learning has explicitly demonstrated the flexibility of this perceptual threshold and how this flexibility can generalise; (5) our ability to image background neurons in the surrounding network has added a further layer of understanding to seminal papers in the field (*Houweling and Brecht, 2008*; *Huber et al., 2008*) and demonstrates that cortical networks largely balance increasing levels of activation with matched suppression.

Nevertheless, it is important to keep in mind caveats inherent to current all-optical approaches that may influence these results, such as limitations in spike readout with calcium indicators (*Chen et al., 2013b*; *Pachitariu et al., 2018*), the photostimulation efficiency of two-photon optogenetic activation (reported here and in *Mardinly et al., 2018*; *Marshel et al., 2019*; *Shemesh et al., 2017*) and the fact that calcium imaging subsamples the full extent of neural activity involved in complex behaviours. It is also likely that our results will be influenced by the exact stimulation parameters, such as stimulus duration, strength, and timing, as has been noted for the detectability of direct cortical activation in the past (*Doron et al., 2014*; *Gill et al., 2020*; *Histed and Maunsell, 2014*; *Huber et al., 2008*). Additionally, as with all studies using trained non-naturalistic behaviour, it is worth considering how training duration might influence our results. We note that animals learn our basic task very quickly and that they continue to quickly generalise learning to new, harder stimuli over time as is often observed in trained sensory paradigms (*Andermann, 2010*; *Gerdjikov et al., 2010*; *O'Connor et al., 2010*). This quick learning means that their performance is stable and effectively saturated by the time we test their perceptual sensitivity, in line with the way in which sensitivity to sensory stimuli is tested in many systems (*Britten et al., 1996*; *Busse et al., 2011*;

*Carandini and Churchland, 2013*; *Morita et al., 2011*). Therefore, our results are interpretable within this standard framework of testing perceptual sensitivity in animals that have learned a non-naturalistic task close to saturation (although see next section), be it contingent on sensory or artificial stimulation. Moreover, as we explore in the next section, we demonstrate that this threshold can change with training, suggesting that the more pertinent feature to consider in relation to perception more generally may be the steepness of animals' psychometric curves. Indeed, our results using artificial ensembles complement the psychometric functions reported in recent studies driving behaviour by optogenetically mimicking sensory ensembles (*Carrillo-Reid et al., 2019*; *Marshel et al., 2019*). Finally, an additional factor influencing our experiments is that bulk 1P optogenetic activation over long training periods may change activity and connectivity patterns, as has been suggested for repeated exposure to 2P optogenetic stimulation of the same neurons over time in the absence of behaviour (*Carrillo-Reid et al., 2016*) and for repeated exposure to the same sensory stimuli over behavioural training (*Chen et al., 2015*; *Khan et al., 2018*; *Peron et al., 2015*; *Poort et al., 2015*; *Wiest et al., 2010*). Future work recording neural responses in the population before, during, and after 1P/2P behavioural training will yield important insight into this process and it will be crucial to compare the changes observed between animals trained on optogenetic stimuli and sensory stimuli.

## The perceptual threshold is plastic and can generalise

We have used our ability to specifically target the same (or different) neurons across multiple days to show that the perceptual threshold is not fixed and depends on learning in a neuron-agnostic manner. This suggests that the perceptual apparatus can flexibly adapt in order to adjust the trade-offs between different kinds of errors while maximising sensitivity (e.g. minimising false positives vs false negatives). It also underscores the brain's ability to reroute its resources (*Chen et al., 2015*; *Hong et al., 2018*; *Huber et al., 2012*; *Kawai et al., 2015*; *Law and Gold, 2008*; *Ölveczky et al., 2011*) to adaptively meet task demands with ever increasing sensitivity and accuracy (*Carandini and Churchland, 2013*; *Fahle, 2005*; *Gilbert et al., 2001*; *Sasaki et al., 2010*). Previous studies addressing this question in the context of sensory tasks have suggested that such learning is associated with changes in the representation of sensory stimuli in primary sensory areas (*Chen et al., 2015*; *Khan et al., 2018*; *Peron et al., 2015*; *Poort et al., 2015*; *Wiest et al., 2010*). However, in our case animals learn despite the stimulus (direct stimulation) being held constant in sensory cortex. We therefore hypothesise that, in our task, such learning-related changes likely occur in downstream regions like S2 (*Chen et al., 2015*; *Kwon et al., 2016*), motor cortex (*Chen et al., 2015*; *Huber et al., 2012*), or striatum (*Sippy et al., 2015*; *Xiong et al., 2015*).

The fact that the learning we observe during the 2P training phase generalises to neurons that are not stimulated during this period suggests that animals can 'learn to learn' (*Harlow, 1949*) within the context of detecting arbitrary cortical activity patterns, similarly to what is observed in tasks relying on more naturalistic neural processing (*Fahle, 2005*; *Rudebeck and Murray, 2011*; *Tse et al., 2007*; *Walton et al., 2010*). Such generalisation of knowledge acquired from one learning epoch to another is a hallmark of the type of powerful statistical learning systems that could underlie some of the brain's most complex, flexible behaviours (*Behrens et al., 2018*; *Eichenbaum and Cohen, 2014*; *Fahle, 2005*; *Gustafson and Daw, 2011*; *Stachenfeld et al., 2017*; *Tolman, 1948*; *Tolman et al., 1946*; *Whittington et al., 2018*; *Whittington et al., 2019*). Our results, and the experimental paradigm that we present, could provide a useful framework to further investigate how learning and credit are assigned to ensembles of neurons in an appropriate yet generalisable way.

The relationship between the perceptual threshold that we measure and how much learning can generalise also merits further investigation. It would be interesting to investigate how nearby stimulated neurons have to be to previously trained neurons, either physically or in terms of tuning similarity, for learning to generalise to them. Indeed another all-optical study hints that generalisation is limited to neurons that share stimulus tuning congruent with ensembles of neurons that have been previously trained (*Marshel et al., 2019*) implying that generalisation is not a universal property. Moreover, it would also be interesting to see whether the strength of generalisation scales with the amount of uncertainty in the preceding trained stimulus set. One could imagine that learning would be more likely to generalise if different neurons were activated on each training day (as in our experiments), or even each trial, than if just one pattern was trained for the same duration. Such volatility in the learning environment is indeed thought to change learning dynamics (*Behrens et al., 2007*;

*Massi et al., 2018*; *McGuire et al., 2014*) and by extension may influence the level of generalisation at the neural level, as has recently been suggested for hippocampal representations (*Plitt and Giocomo, 2019*; *Sanders et al., 2020*).

How might such generalisability of learning arise? It could in part result from increased connectivity between opsin-expressing neurons through plasticity induced during their synchronous activation during 1P training phases, equivalent to how artificial subnetworks might be generated by 2P all-optical methods (*Carrillo-Reid et al., 2016*; *Zhang et al., 2018*, though see also *Alejandre-García et al., 2020* for alternative non-Hebbian mechanisms). In this case, subsequent 2P photostimulation of opsin-expressing neurons on a given day might preferentially recurrently excite other non-targeted opsin-expressing neurons on that day to a greater extent than non-opsin-expressing neurons, as is thought to happen with 2P optogenetic recall of artificially generated subnetworks (*Carrillo-Reid et al., 2016*). This could cause them to become active and 'bound into' the learning process on that day, despite not being directly targeted, and allow them to better drive behaviours when targeted on subsequent days. This would be an intriguing mechanism and, while such changes in connectivity in our task would be 'artificial', other work suggests that similar changes might underlie generalisation in more natural sensory guided tasks. Neurons sharing functional tuning to sensory stimuli tend to form recurrently connected subnetworks (*Carrillo-Reid et al., 2019*; *Chettih and Harvey, 2019*; *Cossell et al., 2015*; *Jennings et al., 2019*; *Ko et al., 2013*; *Marshel et al., 2019*; *Peron et al., 2020*; *Russell et al., 2019*; *Znamenskiy et al., 2018*), which result in non-targeted members being recruited when a subset are targeted for photostimulation (*Carrillo-Reid et al., 2019*; *Jennings et al., 2019*; *Marshel et al., 2019*; *Russell et al., 2019*), and learning can preferentially generalise across neurons within such subnetworks in sensory-guided tasks (*Marshel et al., 2019*). Thus, while the subnetworks that might be generated through our 1P training may be artificial, the process of learning generalisation that we observe may also occur in more naturalistic sensory-driven tasks. Indeed, the fact that this mechanism can extend beyond naturalistic stimuli to aid in detection of arbitrary stimulus patterns speaks to how pivotal it may be in helping the brain generate flexible behaviour.

The combination of sensitivity and flexibility that we report also raises the question of whether animals could be rapidly trained to detect the activity of small numbers of neurons de novo, without prior conditioning. This may be difficult to demonstrate in a realistic experimental timeframe given that animals have a tendency to adopt easy but sub-optimal strategies, such as timing licks to coincide with the mean of the trial-time distribution, when they are faced both with non-naturalistic task design and stimuli that are hard to detect/discriminate. Once these strategies are adopted, such local optima tend to be very hard to train away and are thus better avoided in the first place. Indeed, studies testing perceptual sensitivity to or discrimination of sensory stimuli overwhelmingly begin with easier stimulus types to habituate the animal to the novel task at hand and learning continues as more difficult stimuli are introduced (*Abraham et al., 2004*; *Andermann, 2010*; *Busse et al., 2011*; *Carandini and Churchland, 2013*; *Gerdjikov et al., 2010*; *Histed et al., 2012*; *Lee et al., 2012*; *Morita et al., 2011*; *O'Connor et al., 2010*). These features of sensory-evoked behavioural performance are analogous to how animals in our task constantly adapt over time to reductions in stimulus strength, even down to the lowest stimulus levels tested. Furthermore, all previous studies using all-optical techniques to influence behaviour with cellular resolution optogenetics have incorporated some kind of conditioning phase using either sensory or optogenetic stimuli of progressively lower strength (*Carrillo-Reid et al., 2019*; *Gill et al., 2020*; *Jennings et al., 2019*; *Marshel et al., 2019*; *Russell et al., 2019*) again implying that this may be necessary when probing the limits of perception.

## Perception is sensitive despite matched network suppression

The matched suppression that we observed in the local L2/3 network is in accordance with the general net inhibitory effect of pyramidal neuron stimulation observed in vivo (*Chettih and Harvey, 2019*; *Kwan and Dan, 2012*; *Mateo et al., 2011*; *Russell et al., 2019*) and in detailed network models of cortex (*Cai et al., 2020*). This supports the idea that such networks operate in an inhibition-stabilised regime where one role of inhibition is to control strong recurrent excitation (*Denève and Machens, 2016*; *Murphy and Miller, 2009*; *Ozeki et al., 2009*; *Pehlevan and Sompolinsky, 2014*; *Sanzeni et al., 2020*; *Tsodyks et al., 1997*; *van Vreeswijk and Sompolinsky, 1996*; *Wolf et al., 2014*), although since our perturbations are not targeted to inhibitory neurons with specific tuning

we cannot assess how functionally specific this architecture might be (*Sadeh and Clopath, 2020*). However, these results also seemingly challenge recent work demonstrating that activation of co-tuned ensembles in V1 predominantly activates other similarly tuned neurons in the surrounding network (*Carrillo-Reid et al., 2019*; *Marshel et al., 2019*) and that ablation of some neurons within functional sub-groups reduces activity in the spared neurons (*Peron et al., 2020*). This discrepancy is likely explained by the fact that recurrent excitation is known to increase with tuning similarity such that neurons sharing functional tuning tend to recurrently excite each other (*Cossell et al., 2015*; *Ko et al., 2011*), whereas inhibition is generally less tuned and structured (*Kerlin et al., 2010*; *Bock et al., 2011*; *Fino and Yuste, 2011*; *Hofer et al., 2011*; *Packer and Yuste, 2011*; *Scholl et al., 2015*, though see *Ye et al., 2015*; *Znamenskiy et al., 2018*). Indeed, *Marshel et al., 2019* specifically use a V1 network model relying on recurrent excitation between co-tuned neurons and strong general inhibition, which keeps activity in check, to explain the low threshold and steepness of their psychometric functions. The subset of neurons we targeted, which may not share functional tuning, are unlikely to benefit from such preferential recurrent connectivity but they will likely recruit general inhibition (although, as mentioned above, some form of enhanced connectivity or intrinsic excitability may have been induced during early 1P training). Therefore, the matched suppression and steep psychometric functions that we observe are consistent with this model.

Since it has not been possible up until very recently to assess the impact of such titrated activation of cortical neurons on the local network during behaviour (*Doron et al., 2014*; *Histed and Maunsell, 2014*; *Houweling and Brecht, 2008*; *Huber et al., 2008*; *Tanke et al., 2018*, though see recent work *Gill et al., 2020*; *Marshel et al., 2019*), recent theoretical work inspired by previous behavioural results has explored how simulated neural networks can detect the activation of single neurons (*Bernardi et al., 2020*; *Bernardi and Lindner, 2017*; *Bernardi and Lindner, 2019*). These studies make three key predictions: (1) the pool of readout neurons must be biased in favour of connecting with the stimulated neuron (*Bernardi and Lindner, 2017*), (2) the readout network must include local recurrent inhibition to mitigate noise-inducing neuronal cross-correlations (*Bernardi and Lindner, 2019*), and (3) inhibition must lag excitation in the readout network (*Bernardi et al., 2020*). Since our paradigm allows us to simultaneously monitor the local network response during behavioural detection of similarly sparse activity, we can assess the validity of such predictions in vivo. The strong suppression recruited by our stimulation suggests that powerful inhibition is at work in the network and therefore supports prediction (2). The activity we induce drives behaviour despite this strong local suppression, suggesting that excitation might be transmitted to downstream circuits responsible for driving behaviour before inhibition has a chance to quell it locally. This supports prediction (3). Our results concerning prediction (1) are more mixed. *Bernardi and Lindner, 2017* suggest that the predicted bias could arise due to Hebbian plasticity between stimulated and readout neurons during the initial microstimulation phase of training. They take as evidence for this the fact that naïve animals cannot detect single neurons, something which both we and other similar studies also see (*Carrillo-Reid et al., 2019*; *Gill et al., 2020*; *Histed and Maunsell, 2014*; *Huber et al., 2008*; *Marshel et al., 2019*). The fact that animals generally require initial one-photon priming before being able to detect targeted two-photon stimuli therefore lends some support to prediction (1). However, somewhat contradictory to this prediction is our finding that the amount by which animals improve in their detection of threshold stimuli across sessions is similar irrespective of whether the same or different neurons were stimulated. The bias in connectivity that supposedly develops between target neurons and readout neurons should not extend to other neurons that are not targeted on that day. The fact that we observe such a transfer, manifested in a similar learning rate across different neurons targeted across days, suggests that this bias may be more general than hypothesised above and may apply across most neurons contained within the area where learning has taken place, potentially via recurrent lateral connectivity (either existing or induced during 1P training).

## Outlook

The combination of techniques that we have deployed provide a powerful experimental framework that can be used to test how more nuanced features of cellular identity and specific patterns of cortical activity influence perception. This will bring us closer to the goal of testing precisely which of the candidate features of the neural code underlie the considerable flexibility and processing power of the brain (*Jazayeri and Afraz, 2017*; *Panzeri et al., 2017*).

## Materials and methods

All experimental procedures were carried out under Project Licence 70/14018 (PCC4A4ECE) issued by the UK Home Office in accordance with the UK Animals (Scientific Procedures) Act (1986) and were also subject to local ethical review. All surgical procedures were carried out under isoflurane anaesthesia (5% for induction, 1.5% for maintenance), and every effort was made to minimise suffering.

### Animal preparation

4–6 week old wild-type (C57/BL6) and transgenic GCaMP6s mice (Emx1-Cre;CaMKIIa-tTA;Ai94) of both sexes were used. A calibrated injection pipette (15 µm inner diameter) bevelled to a sharp point was mounted on an oil-filled hydraulic injection system (Harvard apparatus) and front-loaded with virus (either a 1:10 mixture of AAV1-Syn-GCaMP6s-WPRE-SV40 and AAVdj-CaMKIIa-C1V1 (E162T)-TS-P2A-mCherry-WPRE or a 1:8 mixture of AAV1-Syn-GCaMP6s-WPRE-SV40 and either AAV2/9-CaMKII-C1V1(t/t)-mScarlett-Kv2.1 or AAV2/9-CaMKII-C1V1(t/t)-mRuby2-Kv2.1). AAV2/9-CaMKII-C1V1(t/t)-mScarlett-Kv2.1 and AAV2/9-CaMKII-C1V1(t/t)-mRuby2-Kv2.1 virus was diluted in virus buffer solution (20 mM Tris, pH 8.0, 140 mM NaCl, 0.001% Pluronic F-68) 10-fold relative to stock concentration (~6.9 $\times$ $10^{14}$ gc/ml). These constructs were as in *Chettih and Harvey, 2019*. One of the latter two somatically restricted (Kv2.1) opsins was used for all experiments except those targeting the same neurons across days (*Figure 2—figure supplement 3*) where non-restricted opsin was used. Of the 22 mice initially trained on 1P stimulation, 4 were opsin-injected GCaMP6s transgenics and 18 were opsin/indicator-injected WT mice. Of these, 6 mice were used for 2P psychometric curve experiments, 4 of these were opsin-injected GCaMP6s transgenics, and 2 were opsin/indicator-injected WT mice (see *Behavioural training* below for details of mice used for each training phase). Mice were given a peri-operative subcutaneous injection of 0.3 mg/mL buprenorphine hydrochloride (Vetergesic). They were then anaesthetised with isoflurane (5% for induction, 1.5% for maintenance) and the scalp above the dorsal surface of the skull was removed. A metal headplate with a 7 mm diameter circular imaging well was fixed to the skull over right S1 (2 mm posterior and 3.5 mm lateral from bregma) using dental cement. A 3 mm craniotomy was drilled (NSK UK Ltd.) in the centre of the headplate well and the dura removed. Virus was then injected at a depth of 300 µm below the pia either as a single 750 nL injection at 200 nL/min or as ~5 injections of 150 nL virus at 50 nL/min spaced ~300 µm apart. The pipette was left in the brain for 2 min after each injection. Following the final retraction of the injection pipette, a two-tiered 4 mm/3 mm circle/circle chronic window (UQG Optics cover-glass bonded with UV optical cement, NOR-61, Norland Optical Adhesive) was press-fit into the craniotomy, sealed with cyanoacrylate (Vetbond) and fixed in place with dental cement. After surgery, animals were monitored and allowed to recover for at least 6 days during which they received water and food ad libitum.

### Two-photon imaging

For most experiments (*Figure 2*, *3* and *4*) two-photon imaging was performed using a resonant scanning (30 Hz) microscope (Ultima II, Bruker Corporation) driven by PrairieView and a Chameleon Ultra II laser (Coherent). For these experiments a 16x/0.8-NA water-immersion objective (Nikon) and an ETL (Optotune EL-10–30-TC, Gardosoft driver) were used to collect 100 µm *z*-depth imaging volumes (four planes, 33.3 µm spacing) with FOV sizes ranging from 600 $\times$ 600 µm to 850 $\times$ 850 µm (due to ETL magnification changes) at a constant image size of 512 $\times$ 512 pixels and plane-rate of ~7 Hz. The number of neurons recorded in each 2P psychometric curve experiment was 2809 ± 704, *N* = 11 sessions, 6 mice, 1–2 sessions each. For these experiments, we used an orbital nosepiece that allows pitch, roll and yaw to vary in order to get the light-path through the objective orthogonal to the plane of the cranial window to ensure optimal imaging conditions (via the following procedure: https://github.com/llerussell/MONPangle; *Russell, 2020a*). Due to inconsistencies in the window position across days and inaccuracies in the orbital nosepiece positioning and readout, pitch, roll, and yaw can vary significantly across days making it difficult to image the same planes, and thus same neurons, in a volume. We therefore chose to reacquire new volumes each day and target new sets of neurons. For some experiments (some in *Figure 1*, some figure supplements), two-photon imaging was performed using a resonant scanning (30 Hz) microscope (custom-build, Bruker Corporation) driven by PrairieView and a Chameleon Ultra II laser (Coherent). For these

experiments a 25x/0.95-NA water-immersion objective (Leica) and an ETL (Optotune EL-10–30-TC, Optotune driver) were used to collect either 490 × 490 × 100 µm imaging volumes (four planes with ~33 µm spacing) at an image size of 512 × 512 pixels and plane-rate of ~7 Hz or single planes of FOV size 490 × 490 µm and image size 512 × 512 pixels at 30 Hz. GCaMP6s was imaged at 920 nm and mCherry was imaged at 765 nm. For functional volumetric GCaMP6s imaging at 920 nm, the maximum power on sample was 50 mW at the shallowest plane (~130 µm below pia), increasing linearly to 80 mW at the deepest plane (~230 µm) to maintain image quality across the volume (maximum 120 mins total duration per experiment). For single plane GCaMP6s imaging, a maximum power of 50 mW was used for all depths. Power on sample for mCherry/mRuby2/mScarlett (conjugated to C1V1), imaged at 765 nm, was 50–100 mW (maximum 1 min continuous duration).

## Two-photon optogenetic stimulation

Two-photon photostimulation was carried out using a femto-second pulsed laser at 1030 nm (Satsuma, Amplitude Systèmes, 2 MHz rep-rate, 20 W). The single laser beam was split via a reflective spatial light modulator (SLM) (7.68 × 7.68 mm active area, 512 × 512 pixels, optimised for 1064 nm, OverDrive Plus SLM, Meadowlark Optics/Boulder Nonlinear Systems) which was installed in-line of the photostimulation path. Phase masks used to generate beamlet patterns at the focal plane were calculated from photostimulation target *xy* co-ordinates centred on cell-bodies of interest via the weighted Gerchberg-Saxton (GS) algorithm. These targets were weighted according to their location relative to the centre of the SLM's addressable FOV to compensate for the decrease in diffraction efficiency when directing beamlets to peripheral positions. The transformation between SLM co-ordinates and imaging pixel co-ordinates was mapped by burning arbitrary spots in the FOV and calculating an affine transformation between SLM pixel targets and burn targets imaged in 2P. Calibration routines are available at https://github.com/llerussell/SLMTransformMaker3D (*Russell, 2020b*). Spiral patterns were generated by moving all beamlets simultaneously with a pair of galvanometer mirrors. Each spiral consisted of three rotations (i.e. from centre to edge of spiral), 10 µm diameter, 20 ms duration. Powers were adjusted to maintain 6 mW per target neuron. Due to constraints on the total power output of the photostimulation laser to stimulate 200 neurons, we randomly divided all targeted neurons into two groups of 100 which were stimulated as an alternating pair such that each group of 100 neurons was stimulated 10 times at 20 Hz (stimulate first group with 20 ms spiral, 5 ms inter-spiral interval, stimulate second group with 20 ms spiral, 5 ms inter-spiral interval, then return to first group and repeat a further nine times; each group therefore receives 10 × 20 ms spirals with an effective inter-spiral interval, for that group, of 25 ms; ~500 ms total stimulus duration). To stimulate 100 neurons or less all neurons were stimulated simultaneously 10 times at 40 Hz (10 × 20 ms spirals, 5 ms inter-spiral interval, ~250 ms stimulus duration). From our previous work, we expect a single spiral to produce ~1 action potential (*Packer et al., 2015*), thus all 2P stimuli should drive ~10 action potentials. Spiral timing and positioning protocols were generated by a custom MATLAB software suite called Naparm (see *Online design and execution of photostimulation protocols* section of Materials and methods) and executed by the photostimulation modules of the microscope software (PrairieView, Bruker Corporation), the MeadowLark SLM software and our synchronisation software (PackIO; see below).

## One-photon optogenetic stimulation

For Phase 1 of behavioural training an amber LED (590 nm peak wavelength, ThorLabs M590D2) was fixed to a manipulatable arm (DTI clamp, RS components) and press-fit onto the chronic window surface. For subsequent training phases an amber LED (595 nm peak wavelength, ThorLabs M595L3) was mounted in the lightpath above the two-photon microscope objective. LED powers ranged from 0.02 mW to 10 mW. Power and timing of LED photostimulation was controlled by custom MATLAB software through National Instruments data acquisition cards (NI USB-6351, NI USB-6211). All LED stimuli consisted of 5 × 20 ms pulses at 25 Hz of varying powers ranging from 10 to 0.02 mW (measured with PM100A power meter/S130C photodiode sensor, ThorLabs). We did not record activity while animals were learning to detect 1P photostimulation. We thus roughly estimate the spatial extent, number and rate of spiking evoked by pulses of this power, duration, and frequency from the literature as follows. Blue light emitted from an LED shows a ~1 mm HWHM spatial spread in brain tissue and ChR2-based pyramidal neuron activation shows relatively little variation with depth

(*Huber et al., 2008*), although this could be due to activation of superficial processes. C-fos expression also shows that recruitment by optic fibre stimulation of ChR2-expressing pyramidal neurons is limited to a ~0.7 mm$^3$ volume around the fibre tip (*Gradinaru et al., 2009*). Blue and orange light photoactivation of PV interneurons with moderate powers (10 mW) show similar lateral recruitment (~0.5–0.8 mm HWHM) (*Li et al., 2019*), as does orange light Arch-mediate inhibition of pyramidal neurons (~0.5–0.8 mm HWHM) (*Babl et al., 2019*). However, high-power photobleaching indicates that orange light excitation has a two-fold larger lateral extent than blue light and extends across all cortical layers to form a roughly symmetric excitation volume (*Li et al., 2019*), as would be expected from the reduced scattering of red-shifted light in cortical tissue (*Helmchen and Denk, 2005*). Given the fact that we are using an LED instead of a fibre and we are using orange light, we suggest that our excitation volume has a roughly two-fold larger lateral extent than that measured for a blue LED (*Huber et al., 2008*) and for orange light from a fibre/laser (*Babl et al., 2019*; *Li et al., 2019*) and is roughly symmetric. With respect to stimulus frequency, C1V1 should be able to faithfully follow a 25 Hz pulse train with ~90% fidelity, although our stimulus pulses are longer than those used with C1V1 (*Yizhar et al., 2011*) or ChR2 (*Huber et al., 2008*). With respect to pulse duration, for ChR2 a doubling in pulse duration roughly corresponds to a doubling in spike probability (*Histed and Maunsell, 2014*) suggesting that our 10-fold increase in pulse duration relative to the literature (*Yizhar et al., 2011*) might correspond to a ten-fold increase in the number of evoked spikes. Although this might be an overestimate for C1V1 as it has slower channel kinetics than ChR2 (*Yizhar et al., 2011*), it is certainly likely we are driving multiple spikes with each pulse with an upper limit defined by C1V1's inability to reliably follow trains of >50 Hz. These data, in combination with our observation that <25% of neurons express C1V1 (*Figure 1—figure supplement 1e*) and ~50% of C1V1-expressing neurons are photoactivatable, suggest that our initial training powers (10 mW) will drive >5 spikes at between 25 and 50 Hz in ~10% of pyramidal neurons in a 1.5–2 mm$^3$ volume. The lowest powers at the behavioural threshold will likely recruit progressively fewer neurons over smaller cortical volumes with longer spike latencies (*Huber et al., 2008*), and these neurons will fire fewer spikes per pulse with lower reliability (*Histed and Maunsell, 2014*).

## Synchronisation

For synchronisation of imaging frames, photostimulation spirals, sensory stimulation epochs and behavioural trial data during experiments, analogue triggers and waveforms were recorded with National Instruments DAQ cards controlled by PackIO (*Watson et al., 2016*). Behavioural trial timing and licking response contingency analyses were done online by an Arduino Mega microcontroller board controlled by PyBehaviour (see *Behavioural training* section of Materials and methods).

## Online design and execution of photostimulation protocols

To quickly design photostimulation ensembles online during experiments, we used our custom control software, Naparm (https://github.com/llerussell/Naparm; *Russell and Dalgleish, 2020*). Briefly, this software allows users to import images of neuronal populations of interest (i.e. C1V1 expression images) and semi-automatically detect the *xyz* centroids of neuron bodies. These potential target neurons can then be divided into different stimulation groups and the photostimulation protocol defined. Once defined this software saves out all files necessary to synchronise the microscope, lasers, SLM, and master clock software PackIO. For behavioural training experiments, we combined this with our custom two-photon behavioural training software (TPBS) which allows users to import a directory of phasemasks and associate one or more phasemasks with trial types that can be read into and executed by our behavioural control software PyBehaviour (see *Behavioural training* section of Materials and methods).

## Behavioural training

Training began ~21 days post virus injection/window installation. Animals were water-restricted to 85–90% of their pre-training weight throughout the training period and their weight was monitored daily. The majority of each animal's daily water was consumed during behavioural training, topped-up if necessary with additional water post-training. Early one-photon training sessions took place in closed, soundproofed and unlit behavioural training boxes. During training, animals were head-fixed via their headplates and housed in Perspex tubes. A metal reward delivery spout, connected to an

electronic lickometer circuit, was positioned within easy reach of the mouse's tongue to record licks and deliver sugar-water rewards (5 µL, 10% sucrose v/v). Mice were adapted to this procedure over a day or two during which rewards were randomly delivered manually. All subsequent behavioural training was controlled by Arduino-based behavioural control software written in Python (PyBehaviour: https://github.com/llerussell/PyBehaviour; *Russell, 2020c*). This software acted as the master clock, dictating the sequence and timing of behavioural trials, and recording and scoring licking behaviour. The power and temporal characteristics of LED photostimulation patterns were controlled by custom-written MATLAB software and National Instruments hardware (see 'One-photon optogenetic stimulation' section). PyBehaviour controlled the output of this LED software via TTL pulses. At the beginning of each behavioural session, the LED, mounted on a manipulatable arm (DTI clamp, RS components), was press-fit onto the surface of the chronic window. This region was then sealed with tape to minimise direct visual stimulation. Throughout all training sessions and phases, the basic task structure was the same. Trials were triggered after animals withheld licking for $7 \pm 3$ s. Each trial consisted of a response window during which licking behaviour was scored, followed by a 5 s post-stimulus period. The response window lasted for 2 s in training Phase 1 and 1 s in training Phase 2 and 3, although for all behavioural analyses a 1 s window beginning at 0.15 s (0.15–1.15 s) post stimulus was used to exclude unrealistically quick reaction times and ensure consistency of metrics across analyses of different phases (this window is indicated on all behavioural plots). Sessions consisted of 100–400 trials of two types: go trials and catch trials. During go trials, some form of optogenetic stimulus was delivered, and animals were required to lick. Licking was scored as a hit and was rewarded by delivery of a sugar water reward (see above), non-licking was scored as a miss and was unpunished. During catch trials, no stimulus was delivered and animals were required not to lick. Licking was scored as a false alarm, and was unpunished, non-licking was scored as a correct reject and was unrewarded. Neither of these two trial types were cued. Stimulus trials therefore test an animal's detection rate on particular stimuli and catch trials assess an animal's chance response rate. All trial types were pseudorandomly interleaved with a three trial upper limit on consecutive trials of the same type. During the first few sessions, go trials were auto-rewarded 500 ms following the stimulus to encourage learning. As soon as animals reliably began licking in anticipation of the auto-reward, it was turned off for all subsequent sessions. Animals were initially trained on Phase 1 to detect 10 mW. Performance was then manually assessed and the LED power was dropped by half multiple times within a session until animals could detect very low powers (0.05 mW). This usually took ~4 days. Animals then underwent psychometric curve sessions to assess their performance on the lowest LED powers (0.1, 0.08, 0.06, 0.04, 0.02 mW). Animals that could detect 0.1 mW stimuli with d-prime >1 were eligible for transitioning to subsequent experiments (all animals tested achieved this criterion); however, only the subset with clearest expression were transitioned to two-photon training phases to allow for sufficient training time on the all-optical system. These animals were transitioned to Phases 2 and 3 where two-photon stimuli were introduced. For these phases, animals were head-fixed under the microscope on a cylindrical treadmill and the LED was directed through the light-path. Note we expect that animals could be trained from the offset under the microscope using the LED through the objective; however, we opted against this due to time constraints on the all-optical system. An objective well/baffle was used to minimise light leakage. A white-noise mask was played continuously throughout all subsequent training sessions to mask any auditory cues emitted by the galvos during photostimulation. Following selection of neurons for 2P photostimulation (see below) animals began training on Phase 2 where 1P and 2P optogenetic stimuli were interleaved along with catch trials in equal proportions. Initial 2P stimuli targeted 200 neurons and animals required ~1 day to reach good performance. Once animals could reliably detect 200 neurons, we interleaved 2P stimuli targeting a 100 neuron subset of the original 200 neurons. Once animals could reliably detect 100 neurons they were transitioned to Phase 3 where only 2P optogenetic stimulation of those 100 neurons and catch trials were delivered in equal proportions. On some trials in a few Phase 2/3 sessions, auto-rewards were delivered at 0.5, 1 or 1.5 s following the stimulus (depending on the animal's reaction time) to encourage transition to detecting two-photon stimuli. These trials were scored as hits if the animal licked before the auto-reward, otherwise they were conservatively scored as misses (even if they fell within the response window) (see *Behavioural data analysis* below). For the above phases, we used the same FOV across days but did not specifically target the same neurons (see *Two-photon imaging* section of Materials and methods) except in the 'Same' condition in *Figure 2—figure supplement 3*. Following this, a subset of animals were transitioned

to 2P psychometric curve sessions with the remaining subset used for other experiments not reported here (for details, see: https://discovery.ucl.ac.uk/id/eprint/10095170/). During two-photon psychometric curve sessions trial types were pseudorandomly interleaved to ensure an even distribution of trial types across the behavioural session. Trial type ratios were as follows: 15% catch trials, 15% 'easy' 200 neuron trials, 70% trials stimulating smaller numbers of neurons (~12% each for 100, 75, 50, 25, 10, and 5 neurons). All psychometric curve sessions had an initial 10-trial buffer of 'easy' 200 neurons trials to allow the animal to warm up. Multiple FOVs were tested in each animal, but each FOV was tested only once. Mice were eligible for transfer from Phase 2 to Phase 3 and from Phase 3 to psychometric curve sessions once they had achieved d-prime >1 for detecting 2P stimulation of 200 neurons. This did not always happen immediately due to time constraints on the all-optical system. If transition was delayed, in the intervening days mice were often trained to keep them familiar with the task. Summary of *N* for behavioural training: 26 mice did 1P training and of these 18 mice did both high-power and low-power 1P psychometric curves and 4 did just high power and 4 did just low power (these tended to be the best mice which were rushed to later training phases) (*Figure 1*, *Figure 1—figure supplement 3*). From these, 12 mice with the clearest expression were transitioned onto 2P training (*Figure 1*, *Figure 1—figure supplement 5*) and from these 6 mice were used for 2P psychometric curve sessions (*Figures 2–4*) and the remaining 6 were used for other 2P experiments not reported here.

## Image processing

All offline data were analysed using custom software and toolboxes in MATLAB. PackIO data acquired during experiments were used to synchronise imaging frames with photostimulation and behavioural epochs. To avoid the large imaging artefacts caused by 2P photostimulation, we implemented a 'stimulus artefact exclusion epoch' whereby all imaging frames acquired during photostimulation periods were excluded from all processing, analyses and plotting. This is described below. Since it takes 500 ms to stimulate 200 neurons (see *Two-photon optogenetic stimulation* above), and since we wanted the amount of data (volume scans) excluded to be the same for all trial types (to facilitate comparison across them), we need to exclude at least 0–500 ms of imaging data post-stimulus onset on all trials (even though stimuli targeting ≤100 neurons take ~250 ms). In reality, we actually have to remove a slightly longer period of imaging data (five full volumes peristimulus; ~750 ms of imaging data) due to how the photostimulus epoch overlaps with the timecourse of imaging volume acquisition. We explain this below. Photostimulation onsets are not synchronised to begin at the same time as the first plane in a given volume (i.e. they can occur during any plane within a volume). Our volumes contain four planes and if any plane is corrupted by the stimulus artifact then that volume must be discarded. Our imaging frame (plane) rate is 26.8 Hz and, since we acquire four planes per volume, our volume rate is 6.69 Hz. This means it takes ~150 ms to acquire the four planes constituting one volume. Thus, given that the maximum photostimulus duration is ~500 ms (see above), even if photostimulus onset was synchronised to always begin with the first plane of a given volume, the minimum number of volumes we could discard post-stimulus onset would be 4 (3 * 150 = 450, 4 * 150 ms = 600 ms). However, since the photostimulus onset can occur at any time during a volume acquisition, it can also occur during the last plane of the volume immediately preceding the stimulus onset. This volume must also be discarded in addition to the subsequent four volumes which must also be discarded as they will also all contain at least one frame corrupted by the photostimulus. This means that on some trials five volumes will need to be discarded (~750 ms of imaging data). Since we always want to discard the same amount of imaging data across trials to facilitate comparison between them, we have to exclude this maximum value of 5 volumes (~750 ms) peri-stimulus on all trials. Since most reaction times occur before this (560 ± 150 ms reaction time for 2P 200 neuron trials), we are unable to analyse data in the absence of response-related activity (licking, whisking, facial movements). Thus, we had to control for this in our subsequent analyses (see *Neurometric curve analyses* section of *Materials and methods*). Imaging time-series were registered and segmented into ROIs using the Python version of Suite2P (*Pachitariu et al., 2016*). ROIs were manually curated. Neuropil subtracted neuron traces were calculated as:

$$F_{cell} = F_{ROI} - c * F_{neuropil}$$

where the neuropil subtraction coefficient $c$ was estimated separately for each ROI by robust regression between $F_{ROI}$ and $F_{neuropil}$ (*Chen et al., 2013b*). For this estimation process, all photostimulation epochs were excluded and $F_{ROI}$ and $F_{neuropil}$ were downsampled by a factor of 10 (to 0.7 Hz). Coefficients were post-hoc bounded between 0.5 and 1 and any coefficients that could not be reliably estimated were set to the median of all reliably estimated coefficients in that dataset (usually ~0.7). Once $F_{neuropil}$ had been subtracted from $F_{ROI}$, $F_{cell}$ was then re-baselined to the 33rd percentile of $F_{ROI}$ values to ensure accuracy in subsequent $F/F_0$ and $F/\sigma F$ calculations. Neuropil subtraction had a small but significant effect on trial-wise response amplitude; however, we observed a large fraction of negative responses even without neuropil subtraction and response classifications (activated, suppressed and no response) were largely consistent between raw and neuropil subtracted trial-wise responses (*Figure 3—figure supplement 2*). To detect neurons expressing C1V1 offline post experiment (*Figure 1 – figure supplement 1*), we used the Cellpose algorithm followed by manual curation (*Stringer et al., 2020*) using the *cytoplasm* model with diameter = 15 μm. To find Suite2P ROIs that also expressed C1V1, for each Suite2P ROI we copied its spatial footprint to the centroid location of the nearest C1V1 ROI and calculated the overlap between this copied and offset ROI and the original Suite2P ROI. If >50% of pixels overlapped then the Suite2P ROI was considered C1V1[+].

## Neuronal response analysis

For trial-wise analyses, $F_{cell}$ traces were divided into epochs triggered on photostimulation onset for stimulus trials and catch trial response window onset for catch trials (stimulus-triggered averages – STAs). All STAs had a 1 s baseline period. STAs for each trial were converted to $F/\sigma F$ by subtracting the baseline mean from the STA trace ($F$) and dividing the result by the baseline standard deviation ($\sigma F$). Responses on both photostimulation and catch trials were quantified as the average response ~0.7 – 1 s following photostimulus onset so as to avoid photostimulation artefacts (see *Image processing* section above for full description of the stimulus artefact exclusion epoch). For our analyses, we wanted to be able to assess neural responses on single trials as our procedure for mitigating potential lick response artefacts requires trial-wise analyses. Moreover, we cannot assume that the same background neurons will be recruited by photostimulation of a given set of target neurons on each trial due to the high variability and low probability of synaptic transmission (*London et al., 2010*). This therefore precludes the use of a standard statistical test comparing a neuron's stimulus trial response distribution to its catch trial response distribution to assess responsivity. We therefore sought to define activation and suppression thresholds that we could apply to responses on individual trials. We used each neuron's response distribution on correct reject (CR) catch trials, when no stimulus or lick response occurred, to define its separate activation and suppression thresholds (since activation and suppression are readout differently by calcium indicators; see *Otis et al., 2017*; *Figure 2—figure supplement 1b–g*). For each neuron, we defined its activation threshold as:

$$Threshold_{activation} = Mean_{catch} + S.D._{catch}^{\ *} \ Scaling\ Factor_{activation}$$

and its suppression threshold as:

$$Threshold_{suppression} = Mean_{catch} - S.D._{catch}^{\ *} \ Scaling\ Factor_{suppression}$$

where $Mean_{catch}$ and $S.D._{catch}$ are the mean and standard deviation of the distribution of CR catch trial responses respectively (*Figure 2—figure supplement 1b*). We estimated the activation and suppression scaling factors separately for each session (i.e. each session will have a single activation and single suppression scaling factor applied to all neurons), and separately from each other, using a cross-validated empirical procedure where the objective was to ensure that only 5% of neurons were activated and 5% of neurons were suppressed on catch trials, effectively fixing the false positive (FP) rate at 5% for each session (*Figure 2—figure supplement 1c*). For data recorded on each session and for each response type (i.e. in the following example for activation), we swept through a series of potential activation scaling factors (range 1 – 3 in increments of 0.1). At each scaling factor, we ran 10,000 permutations of an 80:20 train:test split over CR catch trials. On each permutation we used the training CR catch trials to define each neuron's activation threshold (using the equation above with the current scaling factor) and then used these thresholds to calculate the proportion of neurons crossing their threshold on each testing CR catch trial and averaged this across trials to get the average proportion activated across testing trials. We then took the median proportion activated

across all permuted train:test splits. We plot this median value for all scaling factors resulting in a distinct curve for each session (*Figure 2—figure supplement 1d,e* data points). We fit each session's curve with cubic interpolation (*Figure 2—figure supplement 1d,e* curves) and then used this to infer the s.d. scaling factor that yields a 5% FP rate across permuted train:test splits for each session and for each response type (*Figure 2—figure supplement 1d,e* box plots). These scaling factors can then be used in conjunction with the full CR catch response distribution as described in the equations above to define an activation threshold and a suppression threshold for each neuron in each session. For subsequent analyses, to quantify the number of target neurons activated we simply calculated the number of neurons in target zones that passed their activation threshold on each trial and averaged this across trials of the same type. To quantify P(Activated) and P(Suppressed) in all neurons (*Figure 3a–c*), we divided the total number of activated or suppressed neurons on each trial, irrespective of whether they were in target zones or not, by the total number of neurons (targets and background) and averaged across trials of the same type. To quantify P(Activated) and P(Suppressed) in background neurons (*Figure 3d,e*), we divided the total number of activated or suppressed background neurons on each trial (target neurons excluded) and divided this by the total number of neurons (targets and background) and averaged across trials of the same type. Note that all neural response data in *Figure 3* and *Figure 4* are hit:miss matched (see below *Hit:miss matching*).

## Classifying target neurons and background neurons

Since the spatial resolution of two-photon photostimulation is not perfect, we defined conservative 3D photostimulation zones around each targeted location in the imaging volume. These had a 10 μm radius, which is ~2 x the lateral resolution of our 2P photostimulation (*Figure 1—figure supplement 2a*), and extended through the entire axial range of the volume (99 μm) (*Figure 2—figure supplement 1a*). Thus, we considered all ROIs with a centroid $\leq$10 μm from any target location to be a potential target neuron, irrespective of their axial displacement relative to the target location. ROIs outside these zones were considered background neurons. Since these exclusion zones are conservative, many neurons fell within them (*Figure 2—figure supplement 1h*), although only a fraction were responsive to photostimulation (*Figure 2—figure supplement 1i,j* 0.18 ± 0.1 fraction activated across neurons within target zones, averaged across trial types, $N$ = 11 sessions, 6 mice, 1–2 sessions each). Nevertheless, this fraction was far above the fraction responsive on catch trials (*Figure 2—figure supplement 1j* stimulus trials: 0.18 ± 0.1 vs catch trials: 0.04 ± 0.01 averaged across trial types p=4.92 $\times$ 10$^{-4}$ paired t-test, $N$ = 11 sessions, 6 mice, 1–2 sessions each) and resulted in numbers of activated target neurons that were 0.46 ± 0.20 times that of the number of target zones (averaged across trial types) and decreased with decreasing number of zones as intended (*Figure 2—figure supplement 1i*).

## Hit:miss matching

We are interested in how the number of target neurons activated modulates responses in the local network (*Figure 3*, *4*). The number of target neurons activated increases the probability of licking (*Figure 2*). Licking itself activates responses in the background network irrespective of photostimulation of target neurons (*Figure 3—figure supplement 1*). Therefore, we sought to minimise the chance that any modulation in background network activity we observe in response to changes in the number of target neurons activated is due to increased recruitment of licking, and lick-evoked activity, by more salient target stimulation. Thus we fixed a 50:50 ratio of hits:misses (lick:no-lick trials) on every trial type (number of target zones) (*Figure 3—figure supplement 1o*). In this way, we can ensure that the proportion of lick corruption in hard to detect stimuli (5 target zones), which mainly result in misses, is similar to that of easy to detect stimuli (100 target zones), which mainly result in hits. Below we describe this procedure using the least salient trial type (5 target zones) of a given session as an example. First, we count the number of trials of this trial type with the minority response type which, in this case, is hits since animals rarely detect 5 target site stimulations. This number is logged as the number of trials to match with trials of the majority response type which, in this case, is misses (as animals often fail to detect 5 target zone stimulations). We then run 100 permutations where we take all the minority trial type trials (hits) and random resamples, of the same number, of the majority trial type trials (misses). For example if there were 2 hits and 15 misses

overall then on each permutation we would take the same 2 hit trials and a random 2 miss trials (sampled from the 15 misses) and calculate our neural metrics across these trials. We then average these metrics across all permutations to get final values reported in network response figures. This procedure was not possible for a few trial types as they only had trials of one response type (i.e. only hits or only misses). These trial types were excluded from analysis.

## Lick analysis

To correlate spontaneous fluorescence traces with spontaneous licking, we removed photostimulation trial periods (0–4 s post-stimulus) and analysed the remaining periods of lick and fluorescence traces spanning the time between the first and last spontaneous lick. We smoothed the lick trace with a Gaussian kernel (sigma = 0.5 s) and calculated the Pearson's correlation coefficient (and associated p-values) between the smoothed lick trace and each neuron's fluorescence trace. To detect spontaneous lick bouts for lick-triggered STA analysis, we excluded all trial periods (0–4 s post-stimulus onset) and found groups of at least three licks that occurred with a minimum 0.5 s inter-lick interval that were separated from preceding licks by at least 1 s.

## Behavioural data analysis

All behavioural trials with $\leq 0.15$ s reaction time were excluded from analysis. Response windows for analysis of all phases extended for 1 s following this period (i.e. 0.15–1.15 s post-stimulus onset). This is indicated in all raster plots. Behavioural trials during imaging experiments which occurred so close to the beginning/end of imaging epochs that the requisite analysis windows were truncated were excluded from analyses. For calculations of reaction time standard deviation we only included sessions where all trial types had responses on >2 trials. For sessions including some auto-rewarded trials (early Phase 1 and Phase 2 sessions), we scored auto-rewarded trials with response times < $t_{auto-reward}$ + 0.15 s as hits and scored auto-rewarded trials with responses after this time, or no response at all, as misses. The basic behavioural response metric used throughout was the $P(response)$ for each stimulation type, calculated as:

$$P(response) = \frac{n_{lick}}{n_{lick} + n_{no\,lick}}$$

where $n_{lick}$ and $n_{no\,lick}$ are the number of trials where the animal licked and did not lick, respectively. For some analyses the catch trial $P(response)$ was subtracted from the $P(response)$ on go trials. This has been indicated where relevant as "catch subtracted". For some behavioural analyses where learning relative to our behavioural response criterion (d-prime >1) was relevant we report performance in terms of d-prime which we computed as:

$$norminv(Hit\ rate) - norminv(False\ alarm\ rate)$$

where $norminv$ is the inverse of the normal cumulative distribution function. We corrected for hit rates/false alarm rates of 1 and 0 according to *Macmillan and Kaplan, 1985*.

## Psychometric curve fitting

We used the MATLAB *psignifit* toolbox (*Schütt et al., 2016a*; *Schütt et al., 2016b*; https://github.com/wichmann-lab/psignifit) to fit log-normal beta-binomial psychometric curves of the following form:

$$\psi(x; m, w, \lambda, \gamma) = \gamma + (1 - \lambda - \gamma)\, S(x; m, w)$$

$$S(x; m, w) = \Phi\left( C\, \frac{\log(x) - m}{w} \right)$$

$$C = \Phi^{-1}(0.95) - \Phi^{-1}(0.05)$$

For which we fit the width ($w$) and threshold ($m$) and fixed the lapse rate ($\lambda$) and guess rate ($\gamma$). $\log$ is the natural logarithm. $\Phi$ and $\Phi^{-1}$ are the cumulative standard normal distribution and its inverse. The lapse rate was fixed as 1 – max(P(Lick)$_{Stim}$) where max(P(Lick)$_{Stim}$) is the maximum response rate

to any stimulus trial type (this was always either 100 or 200 target zone stimulation trials). The guess rate was fixed as the catch trial response rate. Only go (stimulus) trials were used for fitting. For fits aggregating across sessions (*Figure 2d* red curve), the lapse rate and guess rate were fixed using values averaged across sessions. For fits during train-test permutations (*Figure 4d*), the lapse rate and guess rate were fixed at values computed from all data and fixed for all permutations, whereas the width and threshold were re-fit for every permutation.

### Data exclusion criteria

Some animals were removed from analysis of reaction time variability in Phase 1 behavioural sessions as they did not show enough responses in catch trials to compute the mean and standard deviation of reaction time (*Figure 1f, g*). Some trial types for some sessions were excluded from *Figure 3* and *4* as the procedure required to match the number of hit and miss trials resulted in no trials (see *Hit: miss matching* section of the *Materials and methods*). Some trial types were removed from network response analyses (*Figure 3*) as the number of targets activated was <1 on average. No data were removed as outliers. Trials with reaction times ≤0.15 s were excluded from all figures (except for illustrative purposes in *Figure 1—figure supplement 3b*) and analyses as this was deemed to be too quick to be a reaction to the stimulus. In the latter half of each training session if the response probability on the easiest trial type for that session fell below 0.7 (calculated in a 10-trial sliding window centred on each trial) the animal was deemed to be 'sated' and trials beyond this point were excluded from all analyses.

### Statistical procedures

No statistical procedures were used to estimate sample sizes. Appropriate sample sizes were estimated from previous experiments of a similar type. Experiments were not randomised and experimenters were not blinded with respect to experiment and outcome. All data were analysed with custom routines and toolboxes in MATLAB except for psychometric curve fitting which was done using the MATLAB *psignifit* toolbox (see *Psychometric curve fitting* section of Materials and methods). All error bars are given as mean ± s.e.m and all values in the text are mean ± s.d. unless otherwise stated. All boxplots show median and 25th/75th percentile boxes with whiskers extending to the most extreme data not considered outliers (outliers defined as data > $q_3 + 1.5 \times (q_3 - q_1)$ or < $q_3 - 1.5 \times (q_3 - q_1)$ where $q_3$ and $q_1$ are the 75th and 25th percentiles, respectively). Regression coefficients and psychometric function parameters are reported ± 95% confidence intervals where appropriate. Datasets of $N > 8$ were tested for normality with D'Agostino-Pearson's K2 test and analysed according to the result. Datasets of $N \leq 8$ were analysed as nonparametric. Multiple comparisons were corrected using the Bonferroni correction. All tests were two-tailed. Relevant tests are reported in the text.

### Data availability

Import, processing, analysis and figure code is available on Github (*Dalgleish, 2020*; https://github.com/alloptical/Dalgleish-eLife-2020) for use with analysed data (https://doi.org/10.6084/m9.figshare.13135505) and/or unprocessed behavioural session data (https://doi.org/10.6084/m9.figshare.13128950). Raw calcium imaging movies are ~1 TB in size and are thus available upon reasonable request.

## Acknowledgements

We thank Mehmet Fişek for useful discussions about experiments and analysis; Selmaan Chettih and Christopher Harvey for developing and sharing the somatically-restricted C1V1 opsin; Soyon Chun and Agnieszka Jucht for mouse breeding; Sarolta Gabulya and Carmen Fernández Fisac for behavioural training during pilot experiments; and Bruker Corporation for technical support. This work was supported by grants from the Wellcome Trust, Gatsby Charitable Foundation, ERC, MRC and the BBSRC.

## Additional information

### Funding

| Funder | Grant reference number | Author |
|---|---|---|
| European Research Council | 695709 | Michael Häusser |
| Wellcome Trust | 201225 | Michael Häusser |
| Biotechnology and Biological Sciences Research Council | | Michael Häusser |
| Medical Research Council | | Michael Häusser |
| Gatsby Charitable Foundation | | Michael Häusser |

The funders had no role in study design, data collection and interpretation, or the decision to submit the work for publication.

### Author contributions

Henry WP Dalgleish, Conceptualization, Data curation, Software, Formal analysis, Investigation, Methodology, Writing - original draft, Writing - review and editing, HWPD built the behavioural setup, performed surgeries, trained animals, designed and wrote experimental and analysis software, carried out experiments, and analysed data, HWPD, LR, AMP and MH designed the study, HWPD and MH wrote the manuscript with input from LR, AMP and AR; Lloyd E Russell, Conceptualization, Data curation, Software, Formal analysis, Investigation, Methodology, Writing - review and editing, LR performed surgeries, carried out experiments, designed/wrote experimental software, built the microscope and designed and built the behavioural setup, HWPD, LR, AMP and MH designed the study; Adam M Packer, Conceptualization, Supervision, Methodology, Writing - review and editing, AMP designed and built the holographic photostimulation module of the microscope, was crucial in preliminary experiments to optimize the apparatus and behaviour, HWPD, LR, AMP and MH designed the study, MH and AMP supervised the study; Arnd Roth, Conceptualization, Writing - review and editing, AR provided pivotal suggestions for analyses; Oliver M Gauld, Resources, Software, Writing - review and editing, OMG performed surgeries; Francesca Greenstreet, Emmett J Thompson, Investigation, Writing - review and editing, FG and EJT helped design the 1P training paradigm and trained animals on this phase of the experiment; Michael Häusser, Conceptualization, Supervision, Funding acquisition, Writing - original draft, Project administration, Writing - review and editing, HWPD, LR, AMP and MH designed the study, MH and AMP supervised the study, HWPD and MH wrote the manuscript with input from LR, AMP and AR

### Author ORCIDs

Henry WP Dalgleish (iD) https://orcid.org/0000-0002-2390-6361
Lloyd E Russell (iD) https://orcid.org/0000-0001-6332-756X
Adam M Packer (iD) https://orcid.org/0000-0001-5884-794X
Arnd Roth (iD) https://orcid.org/0000-0003-0325-4287
Michael Häusser (iD) https://orcid.org/0000-0002-2673-8957

### Ethics

Animal experimentation: All experimental procedures were carried out under Project Licence 70/14018 (PCC4A4ECE) issued by the UK Home Office in accordance with the UK Animals (Scientific Procedures) Act (1986) and were also subject to local ethical review. All surgical procedures were carried out under isoflurane anaesthesia (5% for induction, 1.5% for maintenance), and every effort was made to minimize suffering.

### Decision letter and Author response

Decision letter https://doi.org/10.7554/eLife.58889.sa1
Author response https://doi.org/10.7554/eLife.58889.sa2

## Additional files

### Supplementary files
• Transparent reporting form

### Data availability

Import, processing, analysis and figure code is available on Github (Dalgleish, 2020; https://github.com/alloptical/Dalgleish-eLife-2020) (copy archived at https://archive.softwareheritage.org/swh:1:rev:5768e890be9fc6428beea16720e488ebd867da67/) for use with analysed data (https://doi.org/10.6084/m9.figshare.13135505) and/or unprocessed behavioural session data (https://doi.org/10.6084/m9.figshare.13128950). Raw calcium imaging movies are ~1TB in size and are thus available upon reasonable request.

The following datasets were generated:

| Author(s) | Year | Dataset title | Dataset URL | Database and Identifier |
|---|---|---|---|---|
| Dalgleish HWP, Russell LE, Gauld OM, Packer AM, Hausser M | 2020 | How many neurons are sufficient for perception of cortical activity? | https://doi.org/10.6084/m9.figshare.13128950 | figshare, 10.6084/m9.figshare.13128950 |
| Dalgleish HWP, Russell LE, Gauld OM, Packer AM, Hausser M | 2020 | How many neurons are sufficient for perception of cortical activity? | https://doi.org/10.6084/m9.figshare.13135505 | figshare, 10.6084/m9.figshare.13135505 |

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
