## [Decision Letter]

**Acceptance summary:**

The study elegantly quantifies the minimal number of neurons that need to be stimulated in order to be behaviorally reported by a mouse. Combination of holographic stimulation methods and of two photon calcium imaging also allows assessing the network impact of targeted stimulations, demonstrating matched activations and inhibitions produced by artificial stimulations of excitatory neurons.

**Decision letter after peer review:**

Thank you for submitting your article "How many neurons are sufficient to generate a percept?" for consideration by *eLife*. Your article has been reviewed by three peer reviewers, including Brice Bathellier as the Reviewing Editor and Reviewer

#1, and the evaluation has been overseen by Ronald Calabrese as the Senior Editor. The following individual involved in review of your submission has agreed to reveal their identity: Luis Carrillo-Reid (Reviewer #2).

The reviewers have discussed the reviews with one another and the Reviewing Editor has drafted this decision to help you prepare a revised submission. While all reviewers recognize that this study brings new important insights about the minimal number of neural activations that can be behaviorally detected, there was a consensus that some of the analyses must be carefully improved and better justified to reach *eLife*'s standards, that the paper must be re-organized and that some of the claims must be toned down.

Summary:

In this manuscript, Dalgleish et al. combined optical imaging and holographic two-photon stimulation with an operant conditioning paradigm to measure how many neurons need to be stimulated in the mouse primary somatosensory cortex to generate a behavioral percept. The techniques used are state of the art and so these findings add precise information about the number and location of stimulated neurons compared to previous studies employing electrical stimulation or 1P optogenetic stimulation. The authors propose a lower bound of ~13 neurons to evoke a 1P optogenetic induced learned behavior. The data are clearly presented, and the conclusions are, for the most part justified, and they can be of interest for the broad audience of *eLife*. However, some conclusions could be more straightforwardly reached than with the present analyses used by the authors, and some important quantifications are not apparent or missing. The important question of confounds between evoked and facial movement related activity is not adequately addressed, and some conclusions (indirect inhibition) appear not to be supported statistically. Detailed prescriptions for revision are given below.

Essential revisions:

1) It would be much clearer to state the results presented in Supplementary Figure 9 right at the beginning. The authors use a complex statistical model in Figure 5 to show that direct activation of only ~40 cells in cortex is sufficient to be detected by the animal at plateau performance. It is confusing to mention that a certain number of neurons are needed to generate a percept to then find out at the end of the Results section that this estimate needs to be decreased by ~ 60% because only ~ 40% of 2P stimulated neurons respond to the stimulation protocol (Supplementary Figure 9). Instead, data from Supplementary Figure 9 should be shown in Figure 2 and the minimum number of cells to be directly activated should be given after correcting for activation probability. The text should be edited accordingly and the question addressed in Figure 5 should be reframed to focus on the question of indirect activations independent of direct activation probability.

2) There seems to be a major problem with Figure 3B-D. The false positive rate of activated or inhibited cells seems to be very high (~12-14%), as seen in catch trials (shadings). This is not surprising given that the authors have a 1 SD statistical threshold which predicts 15% false positive rate. Nevertheless, the authors state that there are inactivated cells although their fraction is below false positive detection rate. The conclusion should be corrected. As far as we can judge from the data there is no statistically detectable indirect inhibition and indirect excitation only becomes significant after 50 targeted cells, and if corrected for false positive rate. This also helps understanding why the statistical model shows no impact of indirect activations.

On the same lines, there are other presentation problems: the scale of Figure 3C makes little sense. It is hard to understand why only 1% of T cell are activated while other cells have a 12% activation rate. The authors should verify and explain why false positive rate is much lower for these cells. Supplementary Figure 6 has no unit and no explanation.

Also the authors better describe spontaneous level. How many neurons are spontaneously active in a given field? How is spontaneous activity related to the artificial activation in terms of the spatial distribution of neurons? The authors should include analyses of the spatial distribution of BG active/suppressed neurons in relation to targeted neurons.

3) Once point 2) is clarified, the only question that remains is whether the indirect activations (those significantly above false positive) are in fact coming from a cortical report of licking. This is indeed possible as they follow exactly licking probability and this is a very interesting point made by this study. However, the authors suggest this can be addressed by subsampling hit trials. Unfortunately, they do not give the rational for this and it did strike the reviewer. To be kept in the paper the procedures used in Figure 4 should be greatly improved and justified. In fact, the statistical model in Figure 5 is more convincing but it should also be better explained.

We also noted that stimulation duration is 250ms and the artefacts are removed over 500ms. The authors should explain why, because measuring DF/F signals just after 250ms (i.e. before reaction time) should solve the issue about whether indirect activations are from licks or from the network.

4) The animals learn in 3 steps, and learning curves for each step are not reported. This should be done so that the reader can appreciate if the animals are at their maximum performance. Moreover, are mice immediately transferring between 1P and 2P in step 2? In Figure 2—figure supplement 3B it is shown that weak stimulation can be learnt to be detected more efficiently. But this is a bit out of context as we do not know how much time it took to learn 100 cells.

The proposal of sparse codification of the responses for this task is contradictory to the fact that the perceptual learning improves with the stimulation of different sets of neurons. It is not clear from the experiments and interpretations how such an exquisite behavioral response with a lower bound of ~13 neurons can generalize to different sets of targeted neurons but cannot be learned by the activation of just few neurons with 2P. In other words, the hypothesis of generalization to arbitrary activity patterns should enable the authors to train the animals with 2P photostimulation?

5) Reaction times are not quantified except in Supplementary Figure 4. They should be included in the main figure and compared to reaction times for very simple tactile detection tasks. E.g. Ollerenshaw et al., 2012, show reaction times in rats that are more rapid (300-250 ms) that those shown in this study. Is cortical stimulation fully comparable to a real perception? Is reaction time dependent on the number of activated cells. Of course, we agree that when there is no significant detection (<25 targeted cells) reaction times should not be computed, as there is no reaction of the animal to the stimulus.

In addition, in Supplementary Figure 4B, reaction times are significantly shorter with the non-soma-tagged C1V1 compared to the soma-tagged variant. How do the authors explain this result? Is it due to non-specific stimulation of fibers of passage, which potentially increases the number of stimulated neurons? If so, shouldn't this also affect the probability of licking?

6) Overall, the experiments demonstrated that the activation of few neurons is sufficient to evoke a learned behavior induced by the bulk activation of many neurons using 1P photostimulation, but even though the authors have the technical capability to describe sensory evoked activity in terms of neuronal ensembles, the authors never relate sensory evoked activity with their targeted neurons. What kind of responses are evoked by 1P 10 mW photostimulation? How many neurons? How many action potentials per neuron? How big is the volume affected by 1P photostimulation? Also, in the first paragraph of the subsection “Very few neurons are sufficient to drive a detectible percept”, during the training protocol with 1P photostimulation the authors decrease the power of stimulation and suggest that the number of neurons activated is decreased. They should show that or consider the possibility that the firing (number of action potentials) is reduced but not the number of neurons active. In particular, in the first paragraph of the subsection “Activation of only a small number of neurons is required to reach perceptual 2 threshold”, the difference between ~13 neurons and ~60 neurons in Huber et al., 2008, could also be due the fact that in the previous paper the number of action potential evoked were controlled whereas in the present study the number of action potentials evoked by 1P or 2P photostimulation are not reported.

7) The title and the impact statement don't reflect the experiments. In particular, the volumetric stimulation with 1P optogenetics doesn't seem to evoke physiological responses. The authors propose that their results prove the lower bound of neurons sufficient to generate a percept, but all the literature cited define a percept as neuronal activity that represents specific features of sensory stimuli. For example, in the references cited hallucinations are defined as "perceptions in the absence of objectively identifiable stimuli. Hallucinations are percepts without corresponding external stimuli". (Corlett et al., 2019). According to such definition the authors are describing hallucinations induced by bulk 1P photostimulation and evoked by 2P activation of few neurons, to which the train the animals to respond. However the relation between neurons responding to 1P stimulation (phase 1 of training) and a sensory evoked percept is missing even though the authors have all the necessary tools to perform such experiments.

So, if the authors want to keep their claims about percepts, they should show the relation of the activity evoked by photostimulation and activity evoked by specific sensory stimuli. In other words, despite that all the references and framework of the manuscript postulate intervention experiments related to percepts, the authors never show the relation between sensory stimuli, targeted optogenetics and behavior. This could be addressed either by toning down the claims, and in particular change the title or demonstrate experimentally that the activity evoked by the targeted activation of neurons represents percepts. For example, a more appropriate title could be: "How many neurons are sufficient to detect direct cortical photostimulation?"

8) Discussion, first paragraph. How the perceptual learning could generalize to the surrounding network? The observed result could be an artifact of a massive network reconfiguration due to 1P photostimulation at early phases of training. The 1P stimulation protocol, which precedes 2P stimulation, may artificially increase connectivity among opsin-expressing neurons and this may explain why the effect of 2P photostimulation on behavior is independent on the identity of stimulated cells (Figure 2—figure supplement 3D). If the network is reconfigured then the animals learned to detect such artificial network or parts of it. Comparing spontaneous activity before and after photostimulation entraining is necessary to clarify this point, and this alternative explanation should be mentioned in the text, especially because the authors seem to favor the opposite scenario, i.e. that "the neurons we targeted probably do no benefit from such recurrent excitation".

9) The main conclusion of the study is that a small number (~ 13) of layer II/III neurons in S1 needs to be stimulated in order to generate a percept. However, this number may strongly depend, among other things, on: i) the number of neurons effectively activated by the 2P stimulation, which is potentially underestimated because of the inability of the calcium indicator to read out small increases in spiking activity under the imaging conditions used in this study; ii) the number of 2P stimulation sessions that animals undergo; iii) the duration and intensity of the 1P stimulation paradigm which may induce different levels of increased connectivity in the stimulated neurons depending on the stimulation conditions; iv) the small number of planes imaged. v) the intensity/duration of the 2P stimulation (e.g., stimulation of 7 neurons with the protocol used in this study may not result in the generation of a behavioral percept, but stronger/longer stimulation of the same 7 neurons might). These issues should be clearly stated in the Discussion, because the authors' estimate on the absolute number of required neurons for generating a percept could strongly depend on their specific experimental conditions and could not necessarily generalize to S1 under any condition.

---

## [Author Response]

Essential revisions:1) It would be much clearer to state the results presented in Supplementary Figure 9 right at the beginning. The authors use a complex statistical model in Figure 5 to show that direct activation of only ~40 cells in cortex is sufficient to be detected by the animal at plateau performance. It is confusing to mention that a certain number of neurons are needed to generate a percept to then find out at the end of the Results section that this estimate needs to be decreased by ~ 60% because only ~ 40% of 2P stimulated neurons respond to the stimulation protocol (Supplementary Figure 9). Instead, data from Supplementary Figure 9 should be shown in Figure 2 and the minimum number of cells to be directly activated should be given after correcting for activation probability. The text should be edited accordingly and the question addressed in Figure 5 should be reframed to focus on the question of indirect activations independent of direct activation probability.

This is an excellent point and we have made the suggested changes. We have clarified our definitions of target and background neurons (Figure 2—figure supplement 1A, Materials and methods “Classifying target neurons and background neurons” subsection) and report the proportion of responsive target neurons at the suggested point both in the text (“Very few cortical neurons are sufficient to drive behaviour” subsection), in the Materials and methods (“Classifying target neurons and background neurons” subsection) and in Figure 2—figure supplement 1H-J. The number of neurons required to drive behaviour is now reported on the basis of the number of target neurons activated in Figure 2 and throughout the manuscript. Figure 4 (which is the equivalent to original Figure 5) now focuses on the contributions of target and background activity to behavioural detection.

2) There seems to be a major problem with Figure 3B-D. The false positive rate of activated or inhibited cells seems to be very high (~12-14%), as seen in catch trials (shadings). This is not surprising given that the authors have a 1 SD statistical threshold which predicts 15% false positive rate. Nevertheless, the authors state that there are inactivated cells although their fraction is below false positive detection rate. The conclusion should be corrected. As far as we can judge from the data there is no statistically detectable indirect inhibition and indirect excitation only becomes significant after 50 targeted cells, and if corrected for false positive rate. This also helps understanding why the statistical model shows no impact of indirect activations.

This comment raises two concerns which we address below: (1) suppression is indefensible, and (2) our thresholds are too lenient.

1) The reviewers are correct in pointing out that there is no suppression beyond that observed on catch trials in the original Figure 3 in which we don’t correct for lick responses (however, we reiterate briefly that there is suppression when we correct for lick responses (original Figure 4D, E, see below)). We apologise for this error in the text and have addressed this through two changes: (i) in the new manuscript we now only reference lick-corrected data (as we feel this is the most accurate representation of responses evoked by photostimulation when behavioural responses are accounted for) meaning that the erroneous passage is no longer included in the manuscript; (ii) inspired by the reviewers’ comments we have implemented more stringent neuronal response thresholds which, while more cleanly separating stimulus-evoked activity from spontaneous activity on catch trials, make almost no qualitative change to our results on lick-corrected data (see point (2) below). This consistency speaks to the robustness of the phenomena that we report. Moreover, we would like to clarify that in the original manuscript our main conclusions concerning suppression were based on the statistically significant suppression that we clearly observed in the background when we correct for lick responses (original Figure 4D) alongside the observation that this scales with the number of target neurons stimulated (original Figure 4E). We believe that this was sufficient evidence for our original claim that we observed network suppression in the original manuscript, though we again apologise for the confusion caused by the description of the original Figure 3.

2) We agree that our neuronal response thresholds were somewhat lenient, but we felt that this was necessary to capture the potentially small responses evoked by synaptic recruitment of background neurons. Our rationale for this was that we expect synaptically recruited responses in the background neurons (particularly activation) to be small and unreliable, due to the features of synaptic connectivity, and we expect suppression to be difficult to detect due to the design of calcium indicators. Moreover, since we wanted to compare the levels of activation and suppression we decided to take thresholds of equal magnitude, but opposite sign, for both. However, as the reviewers point out, this leads to high false positive rates on catch trials that may mask real effects on stimulus trials. While we tried to account for this high false positive rate in the original manuscript by subtracting catch trial response rates from all relevant stimulus trial data (original Figure 4 and 5), as implied by the reviewer, it may be more effective to define more appropriate thresholds.

Therefore, in the revised manuscript we have made these thresholds much more stringent and statistically grounded by empirically estimating them, using cross-validation, from each cell’s distribution of responses on correct reject catch trials (i.e. trials where there should be no signal from photostimulus or licking). This procedure is described in detail in the “Neuronal response analysis” subsection of the Materials and methods and in Figure 2—figure supplement 1. It returns separate activation and suppression thresholds for each neuron, which are defined by the mean and scaling factors for the s.d. of its catch trial correct reject response distribution that yield a 5% false positive rate on catch trials (Figure 2—figure supplement 1B-E). Note that for each experiment we estimate a single activation and single suppression s.d. scaling factor that is applied to all neurons. This procedure is cross-validated in the sense that for each experiment (population of imaged neurons) we sweep through a series of s.d. scaling factors to calculate neuron-wise thresholds on a training subset of catch trials and then calculate the false positive rate (across neurons), using these thresholds, averaged across the remaining testing subset of catch trials. We repeat this for 10,000 train:test splits and infer the s.d. scaling factor that yields a median 5% false positive rate (across neurons, averaged across test catch trials) across all train:test splits. This yields higher s.d. scaling factors than if we simply find the s.d. scaling factor that yields a 5% false positive rate across all catch trials (no cross-validation; Figure 2—figure supplement 1F). These higher cross-validated s.d. scaling factors therefore yield more stringent neuron response thresholds that result in marginally less than 5% false positive rates when used on the full dataset (catch trial response rates in Figure 3, Figure 2—figure supplement 1J).

Despite these more stringent thresholds, the main conclusions of our previous network response analysis, that photostimulation of target neurons recruits matched suppression in background neurons but has little impact on background neuron activation (original Figure 4E), still hold up. We also still observe that background suppression scales with target neuron activation (new Figure 3E), resulting in balanced activation and suppression across all neurons as more target neurons are activated (new Figure 3C), and we still find that background activation does not change (new Figure 3D). Finally, we also still observe that neither background activation nor suppression have any reliable impact on behaviour (Figure 4). The only major qualitative difference we note using these new thresholds is that we now see that photostimulation in general does recruit some background activation (new Figure 3E inset right), though as noted above this is not modulated by the number of activated target neurons (new Figure 3D fit). This is in contrast to our previous observation that photostimulation does not cause activation in the background network (original Figure 4D). As the reviewer points out, this lack of modulation likely underlies the lack of impact of background activation on behaviour.

We thank the reviewers for their excellent suggestion that we adopt more stringent, statistically validated thresholds and we are confident that the qualitive consistency of the large majority of the results between the original and revised manuscripts demonstrates their robustness.

On the same lines, there are other presentation problems: the scale of Figure 3C makes little sense. It is hard to understand why only 1% of T cell are activated while other cells have a 12% activation rate. The authors should verify and explain why false positive rate is much lower for these cells. Supplementary Figure 6 has no unit and no explanation.

We apologise for the confusion in relation to this plot, which is now alleviated by the change in thresholds described above (see our first response to reviewer point 2). Some additional explanation is provided below. In the original version of the manuscript, T neuron activation (proportion of target neurons activated) and BG neuron activation (proportion of background neurons activated) were both defined as proportions using the total number of neurons in the imaging volume as denominators (2000 – 3000 neurons per experiment). This quantification was used to try to describe “what proportion of the population is recruited by direct photostimulation” and “what proportion of the population is recruited indirectly” respectively, and to allow them to be directly compared (since they have the same denominators). However we agree that it is confusing because, despite the fact that photostimulation does cause more robust recruitment of target neurons than background neurons (see below), T neuron activation looks small compared to BG neuron activation because the “maximum possible numerator” is very different. To illustrate this consider the situation where we stimulated 200 target neurons in an imaged volume of 3000 neurons. In this condition the maximum possible numerator for T neuron activation is 200 (if all 200 target neurons responded), whereas for the BG neuron activation it is 2800 (3000 – 200 if all non-targeted neurons responded). Even though it is less likely for background neurons to respond compared to target neurons, this is counteracted by the larger number of neurons we consider in the numerator of BG neurons activated.

We have clarified this in the manuscript and have replaced these acronyms with more interpretable axis labels as follows. BG cell activation and BG cell suppression have been relabelled P(Activated) Background and P(Suppressed) Background respectively and still correspond to the proportion of the population that was activated (or suppressed) but not targeted (Figure 3D, E). We have removed references to T cell activation entirely (since it is confusing) and replaced them with P(Activated) All neurons and P(Suppressed) All neurons which quantify respectively the proportion of all neurons (whole population; targeted and untargeted) that are activated or suppressed (Figure 3A, B). By comparing the P(Activated) and P(Suppressed) in all neurons with the equivalents in the background population one can see that the activation observed in the whole population comes from target neurons (Figure 3A vs. 3D), whereas the suppression is mainly present in the background neurons (Figure 3B vs. 3E). We also point out that we now quantify the proportion of targets activated out of the total number of neurons targeted on each trial type (Figure 1—figure supplement 1J). Comparing all of the above plots it is now clear that the false positive rate is very similar between them (~3 – 5%) and is in line with what one would expect from our threshold estimation procedure (see also our third response to point 2 below). We have also replaced the original Supplementary Figure 6 with the new Figure 3—figure supplement 1A-H for which the units should be more clear.

Also the authors better describe spontaneous level. How many neurons are spontaneously active in a given field? How is spontaneous activity related to the artificial activation in terms of the spatial distribution of neurons? The authors should include analyses of the spatial distribution of BG active/suppressed neurons in relation to targeted neurons.

Spontaneous levels of activation and suppression (quantified on catch trials) are now described as proportions for the whole population (Figure 3A, B), for the background neurons (Figure 3D, E) and for targets (Figure 2—figure supplement 1I, J) and quantified in the text. As expected from our new threshold estimation procedure, ~3 – 5% of neurons are spontaneously active or suppressed on catch trials. This is below the values observed in targets when stimulated (Figure 2—figure supplement 1I, J) and in background when large numbers of targets are stimulated (Figure 3A, B, D, E). We have also described the spatial distribution of activation and suppression in our psychometric curve experiments relative to nearest 2P optogenetic target spot across all neurons (Figure 3—figure supplement 2). Note that this spontaneous activity rate (or “false positive rate”) is slightly lower than 5% because our threshold estimation procedure is cross-validated and the criterion used to infer appropriate thresholds is “median 5% false positive rate across all train:test splits”. This is stringent, resulting in higher response thresholds than when we don’t cross-validate (Figure 2—figure supplement 1F) which will result in marginally lower false positive rates on the full dataset. This is described in more detail above (see our first response to point 2).

3) Once point 2) is clarified, the only question that remains is whether the indirect activations (those significantly above false positive) are in fact coming from a cortical report of licking. This is indeed possible as they follow exactly licking probability and this is a very interesting point made by this study. However, the authors suggest this can be addressed by subsampling hit trials. Unfortunately, they do not give the rational for this and it did strike the reviewer. To be kept in the paper the procedures used in Figure 4 should be greatly improved and justified. In fact, the statistical model in Figure 5 is more convincing but it should also be better explained.

We are glad that the reviewers also appreciate the difficulty of analysing network responses to increasing numbers of activated target neurons when neurons respond to licking directly (Figure 3—figure supplement 1), and licking itself correlates with the number of target neurons activated (Figure 2D). We also agree that lick-related activity in cortex is very interesting and something that subsequent studies need to explore further. We apologize that our rationale for our hit:miss matching procedure wasn’t more clearly explained. We now briefly describe this in the subsection “Suppression in the local network balances target activation”, and detail it in the Materials and methods (“Hit:miss matching” subsection) and in a figure supplement (Figure 3—figure supplement 1O). We re-iterate our rationale here for clarity.

Stimulus trials activating few neurons (e.g. 5) are associated with few hits (licks) and many misses (no licks) whereas stimulus trials activating many neurons (e.g. 200) are associated with many hits (licks) and few misses (no licks). Therefore, when averaging network response metrics across trials, many neuron stimulus trials (200) will have lots of lick-related activity whereas few neuron stimulus trials (5) will have little lick related activity. This means that any modulation of activity in the background network could be a reflection of the increased proportion of trials with lick responses going into the analysis. To address this problem, we fixed the ratio of hit:miss trials (lick:nolick) at 50:50 in our analysis, and kept this consistent across all trial types. For example, let us assume we have 10 trials of each type (10 x 200 neuron stim. trials, 10 x 100 neuron stim. trials etc.). For 5 neuron stimulus trials we have 3 hits and 7 misses whereas for 200 neuron stimulus types we have 7 hits and 3 misses. For 5 neuron stimulus trials, our procedure would take the 3 hit trials and combine them with 100 random 3-trial resamples from the 7 miss trials (to give us 3 hits and 3 misses), calculating the network response metrics on each resample and then averaging across resamples. For the 200 neuron stimulus trials our procedure would take the 3 miss trials and combine them with 100 random 3-trial resamples from the 7 hit trials, calculating the network response metrics on each resample and then averaging across resamples. This procedure therefore ensures that analysis of both 5 and 200 neuron stimulus trials use 3 hits and 3 misses while also attempting to use as much of the data as possible (through the resampling procedure). By doing the same thing for all trial types we remove the variation in the hit:miss ratio with the number of neurons activated. Any residual variation in the background response metrics should therefore be directly due to the number of target neurons activated, and not the more reliable behavioural responses that increasing numbers of target neurons recruit. We hope that, having thus better explained the rationale and method for hit:miss matching that we use, the network results that we report in Figure 3 (the equivalent of the original Figure 4 referenced by the reviewers) are easier to understand.

We also noted that stimulation duration is 250ms and the artefacts are removed over 500ms. The authors should explain why, because measuring DF/F signals just after 250ms (i.e. before reaction time) should solve the issue about whether indirect activations are from licks or from the network.

We apologize for the lack of clarity in our description of 2P photostimulus durations and how they relate to volume acquisition times in the original manuscript. We have now emphasised and clarified the relevant “Two-photon optogenetic stimulation” and “Image processing” subsections of the Materials and methods to explain why we chose our excluded epoch post-stimulus. This is also explained below.

The reviewer is correct in stating that stimuli targeting 100 neurons or less last ~250 ms (we stimulate all neurons simultaneously 10 times at 40 Hz – 10 x 20 ms spirals, 5 ms inter-spiral interval, ~250 ms stimulus duration). However, stimuli targeting 200 neurons last 500 ms because, due to constraints on the total power output of the photostimulation laser (600 mW power on sample; 6 mW per target neuron), we randomly divide the 200 targeted neurons into 2 groups of 100 which we stimulate as an alternating pair (each group of 100 neurons is stimulated 10 times at 20 Hz – stimulate first group with 20 ms spiral, 5 ms inter-spiral interval, stimulate second group with 20 ms spiral, 5 ms inter-spiral interval, then return to first group and repeat a further 9 times; each group therefore receives 10 x 20 ms spirals with an effective inter-spiral interval, for that group, of 25 ms; ~500 ms total stimulus duration). Thus while the majority of stimuli in 2P psychometric curve sessions are 250 ms in duration (≤100 neurons), the strong stimulus trial type is 500 ms (200 neurons). We wanted the analysis period post-photostimulus to be the same across all trial types to facilitate comparison between them and, since the maximum stimulus epoch across all trial types is ~500 ms (for 200 neuron stimulation trials), we have to use this for all trial types, even though ≤100 neuron stimulation trials only have a ~250 ms stimulus epoch. In reality we actually have to remove a slightly longer period of imaging data (5 full volumes peri-stimulus; ~750 ms of imaging data) due to how the photostimulus epoch overlaps with the time-course of imaging volume acquisition (this is explained in detail in the “Image processing” subsection of the Materials and methods).

4) The animals learn in 3 steps, and learning curves for each step are not reported. This should be done so that the reader can appreciate if the animals are at their maximum performance. Moreover, are mice immediately transferring between 1P and 2P in step 2? In Figure 2—figure supplement 3B it is shown that weak stimulation can be learnt to be detected more efficiently. But this is a bit out of context as we do not know how much time it took to learn 100 cells.

We have added supplementary figures detailing the 1P training (Figure 1—figure supplement 3) and the early 2P training (Figure 1—figure supplement 5). We have also added considerable detail describing these phases of the experiment in the text (“Targeted two-photon optogenetic activation of neural ensembles in L2/3 barrel cortex can drive behaviour” subsection) and in the Materials and methods (subsection “Behavioural training”).

From these data it should be clear that mice learn the 1P task very rapidly (often within a single session) and that their performance constantly adapts to reductions in stimulus power (Figure 1—figure supplement 3). This implies that by the end of this phase they are expert and their performance is close to saturation. We have now also referenced this in the Discussion (subsection “The network input-output function for perception is steep”). The amount of time they spend in this phase therefore balances the need to entrain reliable/consistent behaviour with the desire to get to the more relevant 2P phases of the task as quickly as possible without too much behavioural saturation. Mice also very quickly learn to detect 2P photostimulation, again often within the first 1P/2P training session (Figure 1—figure supplement 5A-C), suggesting that our 1P staircase training paradigm was well designed to get mice to the point where they can detect the strongest 2P stimuli that we can deliver, also potentially suggesting that our minimum LED powers are activating a similar number of neurons (~200). Thus since mice are pretty quick to learn 2P photostimulation of hundreds of neurons it is difficult to compare their learning of these stimuli with those of tens of neurons which appear to be much harder for animals to initially detect.

The proposal of sparse codification of the responses for this task is contradictory to the fact that the perceptual learning improves with the stimulation of different sets of neurons. It is not clear from the experiments and interpretations how such an exquisite behavioral response with a lower bound of ~13 neurons can generalize to different sets of targeted neurons but cannot be learned by the activation of just few neurons with 2P. In other words, the hypothesis of generalization to arbitrary activity patterns should enable the authors to train the animals with 2P photostimulation?

As the reviewers point out, the relationship between the lower limit on performance and the extent of learning generalization is indeed fascinating and should be investigated further in subsequent work. We have added sections in the Discussion concerning this issue (subsection “The perceptual threshold is plastic and can generalise”). We also consider whether we would be able to train animals with 2P photostimulation de novo without the 1P priming phase of our experiment, as suggested by the reviewers (see the aforementioned subsection). We believe that this would likely be possible, but only if one initially targeted large numbers of neurons. As in all non-naturalistic paradigms (even those using sensory stimuli) animals must first be primed with stimuli that are orders of magnitude more salient than those that they often later have to detect or discriminate between. This is because animals easily fall into sub-optimal behavioural local minima (i.e. constantly licking, timing licks etc.) if they fail to understand the task. Nevertheless, the priming phase of such studies doesn’t weaken the claim that animals can perceive stimuli with a precision that is defined by the final experiments using the least salient stimuli. We maintain that our experiments can be interpreted in this framework.

5) Reaction times are not quantified except in Supplementary Figure 4. They should be included in the main figure and compared to reaction times for very simple tactile detection tasks. E.g. Ollerenshaw et al., 2012, show reaction times in rats that are more rapid (300-250 ms) that those shown in this study. Is cortical stimulation fully comparable to a real perception? Is reaction time dependent on the number of activated cells. Of course, we agree that when there is no significant detection (<25 targeted cells) reaction times should not be computed, as there is no reaction of the animal to the stimulus.

We have now reported reaction times for all stimuli in all phases of behavioural training (Figure 1F, Figure 1—figure supplements 3D, J, 5D, Figure 2—figure supplement 2A). We do not observe a strong modulation of reaction time by the number of target neurons activated (Figure 2—figure supplement 2A), however we do find that the s.d. of reaction time reduces as more neurons are activated (Figure 2—figure supplement 2B). We have also compared reaction times for optogenetic stimuli (~0.4 – 0.7 s) to those for a variety of whisker stimuli in mice (~0.3 – 0.4 s) in the Discussion (subsection “Activation of only a small number of neurons is required to reach perceptual threshold”), though we have not included the reference suggested by the reviewer as it is in rats and therefore potentially less comparable. We discuss the observation that optogenetic stimuli drive slower behavioural responses and describe some factors that could explain this.

In addition, in Supplementary Figure 4B, reaction times are significantly shorter with the non-soma-tagged C1V1 compared to the soma-tagged variant. How do the authors explain this result? Is it due to non-specific stimulation of fibers of passage, which potentially increases the number of stimulated neurons? If so, shouldn't this also affect the probability of licking?

This is a very astute observation and one that is well worth exploring, as suggested by the reviewer. We have added a section in the Discussion discussing this (subsection “Activation of only a small number of neurons is required to reach perceptual threshold”). We agree that the most likely explanation for the difference in reaction time is stronger, quicker depolarisation generated by the additional opsin in more distal neural processes, as well as recruitment of more distal neurons via processes traversing the photostimulation volume. Moreover it is possible that opsin-induced depolarisation could recruit active dendritic conductances (voltage-gated calcium channels etc.). We agree that the lack of impact on detection rate is somewhat confusing. It’s possible that animals’ performance is saturated at its upper limit (allowing for a lapse rate that is independent of the salience of the stimuli) such that a ceiling effect prevents us from seeing the reduction in detectability that occurs between a very easy to detect stimulus and a slightly less easy to detect stimulus. Moreover if one looks carefully at the distributions in Figure 1—figure supplement 6A one can see that the “weight” of the C1V1-Kv2.1 distribution is lower than that of the regular C1V1 distribution. Thus it is possible that the effect is too small to detect with a dataset of this size.

6) Overall, the experiments demonstrated that the activation of few neurons is sufficient to evoke a learned behavior induced by the bulk activation of many neurons using 1P photostimulation, but even though the authors have the technical capability to describe sensory evoked activity in terms of neuronal ensembles, the authors never relate sensory evoked activity with their targeted neurons.

We agree that this question is fascinating, however we did not collect the data necessary to perform this study. We believe that performing an additional study to address this is beyond the scope of this work. Please also see the response below to major point 7.

What kind of responses are evoked by 1P 10 mW photostimulation? How many neurons? How many action potentials per neuron? How big is the volume affected by 1P photostimulation? Also, in the first paragraph of the subsection “Very few neurons are sufficient to drive a detectible percept”, during the training protocol with 1P photostimulation the authors decrease the power of stimulation and suggest that the number of neurons activated is decreased. They should show that or consider the possibility that the firing (number of action potentials) is reduced but not the number of neurons active. In particular, in the first paragraph of the subsection “Activation of only a small number of neurons is required to reach perceptual 2 threshold”, the difference between ~13 neurons and ~60 neurons in Huber et al., 2008, could also be due the fact that in the previous paper the number of action potential evoked were controlled whereas in the present study the number of action potentials evoked by 1P or 2P photostimulation are not reported.

Since 1P photostimulation was only intended to prime animals to detect 2P photostimulation, and since we do not have a PMT shutter to allow us to do imaging during these sessions, we unfortunately did not record the response to 1P photostimulation. To address this we have added an estimate based on comparisons to previous work in the “One-photon optogenetic stimulation” subsection of the Materials and methods. We expect our 2P photostimuli to elicit ~1 action potential per spiral (Packer et al., 2015), meaning that all 2P trial types should elicit ~10 action potentials. We have noted this in the “Two-photon optogenetic stimulation” subsection of the Materials and methods. We have highlighted in the Discussion that the discrepancy between our numbers and those of Huber et al. could also be partially due to differences in action potential numbers (subsection “Activation of only a small number of neurons is required to reach perceptual threshold”).

However, as we point out in the text, this does not completely account for the difference as Huber *et al.* estimate ~300 action potentials are required to drive behaviour (independent of stimulus pattern, i.e. more action potentials in fewer neurons or fewer action potentials in more neurons), whereas our equivalent estimate would be ~140 action potentials (~10 action potentials in ~14 neurons). We have also added to the Results and Materials and methods the requested additional point that reducing LED power may reduce the probability of firing across a given set of neurons in addition to the number of neurons activated.

7) The title and the impact statement don't reflect the experiments. In particular, the volumetric stimulation with 1P optogenetics doesn't seem to evoke physiological responses. The authors propose that their results prove the lower bound of neurons sufficient to generate a percept, but all the literature cited define a percept as neuronal activity that represents specific features of sensory stimuli. For example, in the references cited hallucinations are defined as "perceptions in the absence of objectively identifiable stimuli. Hallucinations are percepts without corresponding external stimuli". (Corlett et al., 2019). According to such definition the authors are describing hallucinations induced by bulk 1P photostimulation and evoked by 2P activation of few neurons, to which the train the animals to respond. However the relation between neurons responding to 1P stimulation (phase 1 of training) and a sensory evoked percept is missing even though the authors have all the necessary tools to perform such experiments.So, if the authors want to keep their claims about percepts, they should show the relation of the activity evoked by photostimulation and activity evoked by specific sensory stimuli. In other words, despite that all the references and framework of the manuscript postulate intervention experiments related to percepts, the authors never show the relation between sensory stimuli, targeted optogenetics and behavior. This could be addressed either by toning down the claims, and in particular change the title or demonstrate experimentally that the activity evoked by the targeted activation of neurons represents percepts. For example, a more appropriate title could be: "How many neurons are sufficient to detect direct cortical photostimulation?"

We understand the reviewers’ concerns about the connotations of the word “percept”. We have removed it from the text and reworded the title, therefore removing the explicit link between our results, sensory processing and percepts. While we retain our references to false sensory percepts and their role in hallucinations, we now clearly separate our results from the sensory component of these studies, instead clarifying our basic claim that there may be equivalences between the concept of erroneously detecting neural activity in the absence of either sensory or artificial stimuli (subsection “Activation of only a small number of neurons is required to reach perceptual threshold”). We do however continue to use the word “perception” in the context of behavioural detection, in accordance with its widespread use in similar papers driving behaviour with optogenetic and electrical stimulation of barrel cortex and olfactory bulb (Houweling and Brecht, 2008; Huber et al., 2008; Doron et al., 2014; Gill et al., 2020) which demonstrate that it can be used in similar contexts.

With respect to comparing sensory and optogenetic stimuli, we don’t expect our optogenetic stimuli to evoke physiological responses equivalent to sensory stimuli, nor did we intend to claim this. We apologise for this confusion. The original manuscript intended to report the psychometric curve relating “generalised cortical activity” to behaviour and to look at how this influenced the local network. It was not intended to directly relate this to sensory processing, nor do any of our experiments attempt or claim to do so. While this is an interesting question to pursue, we feel that it is beyond the scope of the present study. Furthermore, studies addressing similar issues have already been published by other groups (Carrillo-Reid et al., 2019; Jennings et al., 2019; Marshel et al., 2019; Russell et al., 2019) and we believe that our results offer a set of important complementary insights. We now also reference this comparison in the Discussion (subsection “The network input-output function for perception is steep”).

8) Discussion, first paragraph. How the perceptual learning could generalize to the surrounding network? The observed result could be an artifact of a massive network reconfiguration due to 1P photostimulation at early phases of training. The 1P stimulation protocol, which precedes 2P stimulation, may artificially increase connectivity among opsin-expressing neurons and this may explain why the effect of 2P photostimulation on behavior is independent on the identity of stimulated cells (Figure 2—figure supplement 3D). If the network is reconfigured then the animals learned to detect such artificial network or parts of it. Comparing spontaneous activity before and after photostimulation entraining is necessary to clarify this point, and this alternative explanation should be mentioned in the text, especially because the authors seem to favor the opposite scenario, i.e. that "the neurons we targeted probably do no benefit from such recurrent excitation".

We agree that strengthened recurrent connectivity between opsin expressing neurons induced during 1P training could be a potential explanation for the generalisability of learning. However, since we did not record network activity before and after 1P training it is not possible to explore changes in functional connectivity as suggested. We have now highlighted this in the Discussion (subsection “The network input-output function for perception is steep”), also emphasising in detail the interpretation of our generalisability result that the reviewers mention and exploring how it relates to other similar studies using more naturalistic sensory stimuli (subsection “The perceptual threshold is plastic and can generalise”). We have also added a caveat to this effect to the new version of the sentence referred to by the reviewers (subsection “Perception is sensitive despite matched network suppression”). We don’t necessarily see this interpretation as problematic since studies driving behaviour with sensory stimuli/optogenetic stimuli mimicking sensory stimuli often cite such recurrent connectivity as a mechanism for improving performance in behavioural tasks (Carrillo-Reid et al., 2019; Jennings et al., 2019; Marshel et al., 2019; Russell et al., 2019) and indeed one has explicitly suggested that such recurrence allows rapid transfer learning between neurons sharing functional tuning with neurons that have previously been trained (Marshel et al., 2019). This idea is therefore not specific to the artificial stimuli that we deliver, though as pointed out by the reviewers in our hands it would be set up by artificial generation of neural ensembles (Carrillo-Reid et al., 2016; Zhang et al., 2018). Indeed we favour this interpretation in the revised manuscript as it would offer a functionally grounded account of how the similarity between stimuli, and the ensembles that encode them, could facilitate rapid transfer of learning. We have also cited an intriguing recent preprint suggesting that repeated optogenetic stimulation of populations of opsin-expressing neurons does not actually increase the strength of recurrent connections between them (Alejandre-García et al., 2020), instead resulting in an increase in intrinsic excitability that unmasks existing weak synaptic connections. This suggests that 1P optogenetic training may not induce the formation of new connections between opsin expressing neurons, but would have to rely on existing connections. Given that during the 2P psychometric curve experiments (when we test for changes in the detectability of small numbers of neurons) we move FOVs by several hundred microns between training days it is likely that the probability of existing excitatory connectivity between the random sets of neurons targeted across days would be quite low. Nevertheless, such subtle intrinsic excitability-induced unmasking of weak synapses could also mediate a recurrent excitation-based explanation for the transfer learning that we observe.

9) The main conclusion of the study is that a small number (~ 13) of layer II/III neurons in S1 needs to be stimulated in order to generate a percept. However, this number may strongly depend, among other things, on: i) the number of neurons effectively activated by the 2P stimulation, which is potentially underestimated because of the inability of the calcium indicator to read out small increases in spiking activity under the imaging conditions used in this study; ii) the number of 2P stimulation sessions that animals undergo; iii) the duration and intensity of the 1P stimulation paradigm which may induce different levels of increased connectivity in the stimulated neurons depending on the stimulation conditions; iv) the small number of planes imaged. v) the intensity/duration of the 2P stimulation (e.g., stimulation of 7 neurons with the protocol used in this study may not result in the generation of a behavioral percept, but stronger/longer stimulation of the same 7 neurons might). These issues should be clearly stated in the Discussion, because the authors' estimate on the absolute number of required neurons for generating a percept could strongly depend on their specific experimental conditions and could not necessarily generalize to S1 under any condition.

We completely agree, and we have explored this in detail in the Discussion: (subsection “The network input-output function for perception is steep”).